

# The Process and Value of Reprogramming a Legacy Global Hydrological Model

Emmanuel Nyenah[1,2], Petra Döll[1,2], Martina Flörke[3], Leon Mühlenbruch[3], Lasse Nissen[1] and Robert Reinecke[4]

[1]Institute of Physical Geography, Goethe-University Frankfurt, 60438 Frankfurt am Main, Germany
[2]Senckenberg Biodiversity and Climate Research Centre (SBiK-F), 60438 Frankfurt am Main, Germany
[3]Institute of Engineering Hydrology and Water Resources Management, Ruhr University Bochum, 44801, Bochum, Germany
[4]Institute of Geography, Johannes Gutenberg-University Mainz, 55128 Mainz, Germany

*Correspondence to*: Emmanuel Nyenah (Nyenah@em.uni-frankfurt.de)

## Abstract

Global hydrological models (GHMs) improve our understanding of water flows and storage on the continents and have undergone significant advancements in process representation over the past four decades. However, as research questions and GHMs become increasingly complex, maintaining and enhancing existing model codes efficiently has become challenging. Issues such as non-modular design, inconsistent variable naming, insufficient documentation, lack of automated software testing suites and containerization hinder the sustainability of GHM research software as well as the reproducibility of study results obtained with the help of GHMs. Although some GHMs have been reprogrammed to address these challenges, publications focus on evaluating model performance and do not describe the reprogramming process. To support the reprogramming of large geoscientific research software, we present in detail how the GHM WaterGAP was reprogrammed into sustainable research software. Following an agile project management approach, the software was rewritten from scratch in Python with a modular Model-View-Controller architecture, including development practices such as open-source licensing, version control, unit testing, linting, containerization, consistent and meaningful variable naming, and comprehensive in-code and external documentation. Due to the switch from C/C++ in the legacy code to Python, execution time doubled. Our evaluation of the reprogrammed WaterGAP code against software sustainability criteria and FAIR4RS principles indicates that the reprogramming substantially improved the software usability, maintainability, and extensibility, making the reprogrammed WaterGAP software much more sustainable than its predecessor. The new WaterGAP software can be easily understood, applied and enhanced by novice and experienced modelers and is suited for collaborative code development across diverse teams and locations, fostering the establishment of a community GHM.



## 1 Introduction

Over the past four decades, global hydrological models (GHMs) have made remarkable progress in process representation, such as the incorporation of artificial reservoirs and the differentiation of groundwater and surface water use (Telteu et al., 2021). While the most widely used spatial resolution of GHMs is still 30 arc-minutes, the demand for more spatially resolved information has led to GHMs running at 5 arc-minutes (Eisner, 2016; Flörke et al., 2018; Sutanudjaja et al., 2018), 3 arc-minutes (Choulga et al., 2024) or even 30 arc-seconds (Hoch et al., 2023). Still, further progress is required to improve GHMs, e.g., to better represent human-environment interactions and reduce model uncertainties by improved integration of model output observations (Burt and McDonnell, 2015; Döll et al., 2024).

As research questions and, thus, GHMs become more complex, maintaining and further developing an existing model code in an efficient manner becomes increasingly challenging. Similar to other research software, GHMs are developed by scientists with limited software development training, time, and funding, and thus lack the software quality that is required for sustainable research software (Döll et al., 2023). A recent assessment of the software sustainability of global impact models, including nine GHMs, revealed limited accessibility, low adoption of containerized solutions, non-modular design, suboptimal comment density (defined as the number of lines of comment per total lines of code), and the absence of software testing (Nyenah et al., 2024). In addition, poor software quality also hinders the reproducibility of computational research (Döll et al., 2023; Reinecke et al., 2022).

While there are various definitions of sustainable research software (e.g. Anzt et al., 2021; Katz, 2022; Venters et al., 2018), the definition by Anzt et al. (2021) provides clear and measurable qualities that are suitable for evaluating the sustainability of complex research software such as a GHM. Anzt et al. (2021) define sustainable research software as software that 1) is maintainable, extensible, and flexible, i.e., adapts to user requirements, 2) has a defined software architecture, 3) is testable thus ensuring software components function as intended through practices like unit testing, 4) has comprehensive in-code and external documentation, and 5) is freely accessible, i.e., licensed as open source with a digital object identifier (DOI) for proper attribution. In the following, the term "research software" includes the algorithms, source code files, computational workflows, and executables developed during the research process or for a research objective (Barker et al., 2022).

The sustainability of GHMs and the reproducibility of GHM-based research could be significantly enhanced by reprogramming GHMs using modern best practices (Nyenah et al., 2024). These include adopting an open-source license, containerization, implementing a modular architecture, selecting informative variable names, and improving both the density of comments within the code and external documentation (refers to manuals, guides, tutorials, and any materials that provide information about your software to users and developers) (Nyenah et al., 2024). Additionally, applying FAIR (Findable, Accessible, Interoperable, and Reusable) principles for research software (FAIR4RS) improves research software reusability,



reproducibility as well as transparency (Barker et al., 2022). For instance, the eWaterCycle (Hut et al., 2022) platform has taken initial steps toward implementing FAIR principles for hydrological models, enabling other researchers to use these

models without significant support from the original authors. To improve comprehension, usage, maintenance, extension, and collaborative development, the global land surface model CLASSIC (Melton et al., 2020) and the GHMs such as HydroPy (Stacke and Hagemann, 2021) and PCR-GLOBWB (Sutanudjaja et al., 2018) were reprogrammed. However, the publications of these reprogrammed software typically focus on evaluating model performance and lack a detailed account of the reprogramming process.

To address this gap and facilitate the reprogramming of other legacy research software, we present the reprogramming process to transform the GHM WaterGAP into a sustainable research software in a detailed and comprehensive way. Key aspects of the reprogramming process include software requirement specifications, project management methods, use of version control, in-code and external documentation, peer code review and other measures to maintain code quality such  software testing, and

linting (analyzing source code for programming errors and suspicious constructs, (Heričko and Šumak, 2023)). In the so-called ReWaterGAP project, a new code that implements all algorithms and functionalities of the most recent WaterGAP model version 2.2e (Müller Schmied et al., 2024) was written from scratch, to replace a legacy code whose development started in 1996. Improvements include a modular architecture, a modern programming language (Python), and state-of-the-art practices such as version control, open-source licensing, consistent variable naming, comprehensive documentation, and software

testing; while maintaining computational performance. The major aim of reprogramming was to facilitate efficient and joint code development across multiple locations and by various developer groups to establish a community global hydrological model that can be easily understood. In addition, we wanted to ensure that all model outputs and publication results are reproducible (Döll et al., 2023; Hutton et al., 2016; Reinecke et al., 2022).

The paper is structured as follows: section 2 introduces the WaterGAP model and the legacy software. Sustainability criteria for research software and methods relevant to this study are presented in section 3. After describing the reprogramming process in section 4, we present the architecture and the new features of the reprogrammed software in section 5. In section 6, we evaluate the new WaterGAP software against selected sustainability criteria and the FAIR4RS principles. We also demonstrate that the reprogrammed and the legacy software result in very similar model output (section 7) and present user perspectives

on the reprogrammed software obtained by an online survey (section 8). The discussion in section 9 is followed by our conclusions.



## 2 The WaterGAP model

### 2.1 Model description

WaterGAP is a global-scale water resources and use simulation model that has been developed since 1996 (Müller Schmied et al., 2024). Covering all land areas of the globe except Antarctica, it has been widely used in studies of water scarcity, drought, and ecologically relevant streamflow characteristics, considering the impacts of human water use, reservoirs, and climate change (Müller Schmied et al., 2021) as well as inter-basin transfers (Flörke et al., 2018). Model results have contributed to various reports by the Intergovernmental Panel on Climate Change (IPCC) (Jiménez Cisneros et al., 2014) and the State of Global Water Resources Report by the World Meteorological Organization (WMO) (WMO, 2024). WaterGAP is also a key participant in the Inter-Sectoral Impact Model Intercomparison Project (ISIMIP) where the focus is on both model evaluation (and or improvement) and the impact assessment of anthropogenic changes such as human water use or climate change (Frieler and Vega, 2019; Heinicke et al., 2024; Warszawski et al., 2014). WaterGAP output is utilized in diverse research fields, e.g., life-cycle assessment (Boulay et al., 2015; Schomberg et al., 2021) or freshwater ecology (Datry et al., 2021; Domisch et al., 2017; Schneider et al., 2017). Furthermore, simulated water storage anomalies were used by geodesists to evaluate GRACE (Gravity Recovery and Climate Experiment) satellite observations of Earth's dynamic gravity field (Kusche et al., 2009; Schmidt et al., 2006) and streamflow estimates were used to analyze thermal and hydropower production (van Vliet et al., 2016; Wan et al., 2022).

WaterGAP exists as two main model families distinguished by their spatial resolutions. WaterGAP 2 operates at a 30 arc-minutes resolution, with the latest version being 2.2e (Müller Schmied et al., 2024), while WaterGAP 3 uses a finer 5 arc-minutes resolution (Flörke et al., 2018). WaterGAP 2 and 3 also differ with respect to some algorithms and input data. WaterGAP consists of five sectoral water use models for irrigation, livestock, domestic, manufacturing, and cooling of thermal power plants. These water use models are interlinked through the Groundwater Surface Water Use (GWSWUSE), which computes potential net abstractions from groundwater and surface water based on the output of the water use models. These net abstractions are an input to the WaterGAP Global Hydrology Model (WGHM) (Müller Schmied et al., 2021, 2024). All models combined are referred to as the WaterGAP model. WGHM computes both vertical water balance encompassing the canopy, snow, and soil components and lateral water balance, which includes groundwater, lakes, artificial reservoirs, wetlands, and rivers. The basin-specific calibration of WGHM distinguishes WaterGAP from other global hydrological models (Müller Schmied et al., 2021). It aims at reducing the bias of simulated streamflow by using a simple method (see Section 5.2 of Müller Schmied et al., 2024) to match the long-term mean annual observed streamflow at 1509 basin outlets, covering 55% of the global drainage area (excluding Antarctica and Greenland). This paper only concerns the reprogramming the GWSWUSE and WGHM models operating at the 30 arc-minutes resolution.





## 2.2 Characteristics of the legacy software

The legacy software of WaterGAP was primarily written in C and C++ by PhD students and postdoctoral researchers with diverse programming backgrounds. The software is hosted on a private GitHub repository under the GNU Lesser General Public License (LGPL v3.0). The available model documentation includes two model description papers (Müller Schmied et

al., 2021, 2024), as well as a number of documents not available to the public. The latest WGHM version (WaterGAP 2.2e) is archived on Zenodo (https://zenodo.org/records/10026943). However, this version is limited to review only and cannot be executed by external users. The source code of the latest WGHM version contains 25,204 lines of code across 85 files. The linking model GWSWUSE contains 3550 lines of code across 14 files.

In addition to lacking comprehensive and easily accessible external documentation, the sustainability of the legacy software is constrained by several software characteristics. The software has a limited modular structure, consisting of a collection of "script-like" files, with some having up to 6,000 total lines of code (Nyenah et al., 2024). The WGHM code has a non-optimal comment density (approximately 25%) compared to the recommended 30–60% (Nyenah et al., 2024). This makes the model code challenging to read and maintain. In addition, the WaterGAP software uses a non-standard binary file format called UNF

for input and output data instead of the now widely used NetCDF format. Climate forcing data, for example, must first be converted to UNF before use. This introduces additional complexity, potentially creating barriers and susceptibility to errors for external users unfamiliar with the format or conversion tools. No unit tests (verifying that individual code components work correctly) exist to check if algorithms produce outputs within acceptable ranges. Also, WaterGAP does not utilize containerization technology, which makes the reproduction of research results more difficult.

## 3 Methods

### 3.1 Software evaluation against sustainability criteria and FAIR4RS principles

We examined nine indicators for research software sustainability evaluated in the study of Nyenah et al. (2024; their Table 1), consisting of five indicators of best practices in software engineering and four indicators of source code quality. Additionally, we assessed the research software against the FAIR4RS principles outlined by Barker et al., (2022). The FAIR4RS principles

aim to enhance transparency, reproducibility, and reusability of research software (Barker et al., 2022; Chue Hong et al., 2022). Sustainability overlaps with the FAIR4RS principles, particularly regarding reusability, which ensures that software can be understood, modified, built upon, or incorporated into other software. Findability and accessibility also support sustainability by enabling users to easily retrieve the software.

We describe the sustainability indicators in sections 3.1.1, 3.1.2 and the evaluation of the FAIR4RS principles in section 3.1.3.






### 3.1.1 Indicators for best practices in software engineering

*External documentation*. Effective use and ease of software maintainability relies on clear and extensive external documentation (Nyenah et al., 2024; Wilson et al., 2014). Here, we evaluate the availability and extensiveness of external
documentation by analyzing the following components:

- *Installation guide:* step-by-step instructions for installing the software.
- *Tutorials*: practical guides for running various experimental setups (e.g., running WGHM for a specific basin).
- *User guide*: detailed information tailored to users with varying levels of programming expertise, from beginner to advanced software users .
- *Reference guide*: in-depth descriptions of the  model processes and the governing equations.
- *Contributor guide and frequently asked questions (FAQs)*: guidelines for contributing to the code, reporting documentation typos or software bugs, and resolving common issues.
- *Glossary*: definitions of terms and acronyms (e.g., variable names) used in the documentation to enhance user comprehension.


*Version control and automation*. Version control systems play a crucial role in software development by facilitating change tracking, enabling collaborative work, and maintaining a comprehensive history of software evolution (Wilson et al., 2014). We evaluate the use of version control systems and examine specific version control practices. One key aspect we consider is the choice between public and private repositories, significantly affecting the repository's transparency and accessibility.
Regarding automation practices integrated into the repository, we consider automated testing, linting, and documentation. Such automation ensures consistent testing of newly added or modified code, adherence to coding standards, and up-to-date documentation throughout the development lifecycle.

*Use of an open-source license*. Following the approach outlined by Nyenah et al. (2024), we determine the presence of open-
source licenses by reviewing license files within repositories and comparing them with licenses approved by the Open Source Initiative (OSI) ([https://opensource.org/licenses](https://opensource.org/licenses)) (Nyenah et al., 2024).

*Number of active developers.* The number of active developers indicates the capacity for ongoing software development and maintenance (Nyenah et al., 2024). We measured this by counting individuals who made commits to the  codebase of the
legacy and the reprogrammed code within the past two years (2023–2024)..

*Containerization.* Containerization is a method of packaging software applications along with their entire runtime environment (including libraries) into isolated units called containers (Nüst et al., 2020). Traditionally, reproducing experiments is challenging due to variations in for instance operating systems and installed libraries across different machines.



Containerization solves this by encapsulating the complete and specific software environment within a container image. This image becomes a portable and immutable snapshot of the execution environment. When an experiment is executed in a container derived from a specific image, the researcher can be confident that the environment is exactly the same, regardless of the host system. This guaranteed consistency across different machines and over time is the key reason why containerization strongly supports the reproducibility of research software experiments. Here we only evaluate whether a containerization

solution exists or not.

### 3.1.2 Indicators for source code quality

*Public availability of an automated testing suite*. We adopted the approach proposed by Nyenah et al. (2024), using the availability of an automated testing suite as a proxy for evaluating the ease of testing software functionality. The actual property

of interest is test coverage, which verifies the software's functionality. However, existing test coverage tools do not support Python functions with Numba decorators. Numba is a Just-in-Time compiler that boosts performance by converting Python functions into machine code, and the decorators specify which functions Numba should compile into machine code at runtime (Lam et al., 2015). See example issues reported on GitHub (https://github.com/nedbat/coveragepy/issues/849) and Stack Overflow (https://stackoverflow.com/questions/26875191/analyzing-coverage-of-numba-wrapped-functions).


*Compliance with coding standards.* Coding standards represent a set of industry-acknowledged best practices that provide guidelines for software code development (Wang et al., 2008). In our analysis, we employed two distinct tools to evaluate the adherence to coding standards for the legacy and the reprogrammed code. For the legacy code, we utilized the CLion static analysis tool (https://www.jetbrains.com/help/clion/code-inspection.html), which assesses code against the C/C++ Core

Guidelines. This tool identifies potential issues in the source code, including errors, typos, and warnings. However, it does not provide an aggregate score, which makes it difficult to interpret the results. Nevertheless, a high number of identified issues typically indicates critical problems that could compromise software reliability or maintainability. For the reprogrammed code, we employed Pylint (https://pylint.readthedocs.io/en/latest/index.html), which assesses compliance with PEP-8 coding conventions. Pylint quantifies adherence to this coding standard by assigning a maximum score of 10 for perfect compliance,

with no lower bound (Molnar et al., 2020; Nyenah et al., 2024).

*Comment density*. We compute comment density as the ratio of the number of lines of comments to the total lines of code (TLOC). TLOC refers to the sum of source lines of code (SLOC) and comment lines. SLOC, in turn, represents the non-blank, non-comment lines within a source file. We regard a comment density of 30% to 60% as optimal (Arafat and Riehle, 2009;

He, 2019; Nyenah et al., 2024).





*Modularity.* We evaluate the modularity of the software by the TLOC per file metric, with an ideal range of 10 to 1,000 TLOC per file (Nyenah et al., 2024). This metric reflects the organization of source codes into smaller, manageable modules, each focusing on a specific functionality. Modules within this range are typically easier to read, modify, and reuse.


### 3.1.3 Evaluation against FAIR4RS principles

Based on Barker et al. (2022), the reprogrammed software were evaluated against eleven main principles. For findability (F), we verified whether the software is assigned a globally unique and persistent identifier (F1) and described with rich metadata (F2). We aimed at ensuring that the metadata included the software identifier (F3) as well as is searchable and indexable (F4).

Regarding accessibility (A), we checked whether the software is retrievable via a standardized protocol (A1) and the metadata remain accessible even if the software was to become unavailable (A2). To evaluate interoperability (I), we examined whether the software reads, writes, and exchanges data following domain-relevant standards (I1) and includes qualified references to other objects (I2). For reusability (R), we checked whether the software is given a clear and accessible license (R1), includes qualified references to other software (R2), and meets community standards (R3).

**3.2 Differences between the outputs of the reprogrammed and legacy software**

To verify that the reprogrammed software computes the same model output as the legacy software unless an algorithm was changed to improve the computation, we compared the globally averaged water balance components and the global maps of renewable water resources, i.e. the long-term differences between precipitation and actual evapotranspiration in the 30 arc-minute grid cells (Müller Schmied et al., 2021). Both were calculated using the WGHM output of the legacy and the

reprogrammed software. It is important to note that WGHM output also reflects the difference in GWSWUSE output as the potential net abstractions computed by GWSWUSE are incorporated into WGHM. The model setup for calculating the water balance components was "anthropogenic" (i.e., considering human water use and artificial reservoir management), with a 5-year spin-up and a simulation period from 1901-2019. The water balance analysis focuses on key water balance components and the long-term average volume balance error for five distinct periods: 1961–1990, 1971–2000, 1981–2010, 1991–2019,

and 2001–2019. Results for the legacy code are provided in Table 4 of Müller Schmied et al. (2024).

Renewable water resources were calculated over the period 1981-2010 by running the model in the naturalized mode, i.e., without considering human water use and reservoir operations, as detailed in sections 4.7.3 and 7.2.1 of Müller Schmied et al. (2021). Notably, total water resources can have negative values if evapotranspiration in a cell is large than precipitation due to inflow from upstream cells. Output differences for many other model output variables were checked during the reprogramming

process but are omitted here.



### 3.3 User survey

We conducted an online survey to determine perspectives on the reprogrammed WGHM software, focusing on the research software sustainability (see also the complete questionnaire and answers in the Supplement). The survey evaluated the readability, comprehensibility, modifiability, and documentation quality of a code snippet that implements the Priestley-Taylor

potential evapotranspiration algorithm in the reprogrammed software. The survey was conducted within approximately one month, receiving 217 clicks, with 64 participants completing it.

### 4 The reprogramming process

The reprogramming process for WaterGAP was enabled by a grant that financed one full-time PhD and a student assistant over three years. Figure 1 shows the timeline of the reprogramming process.


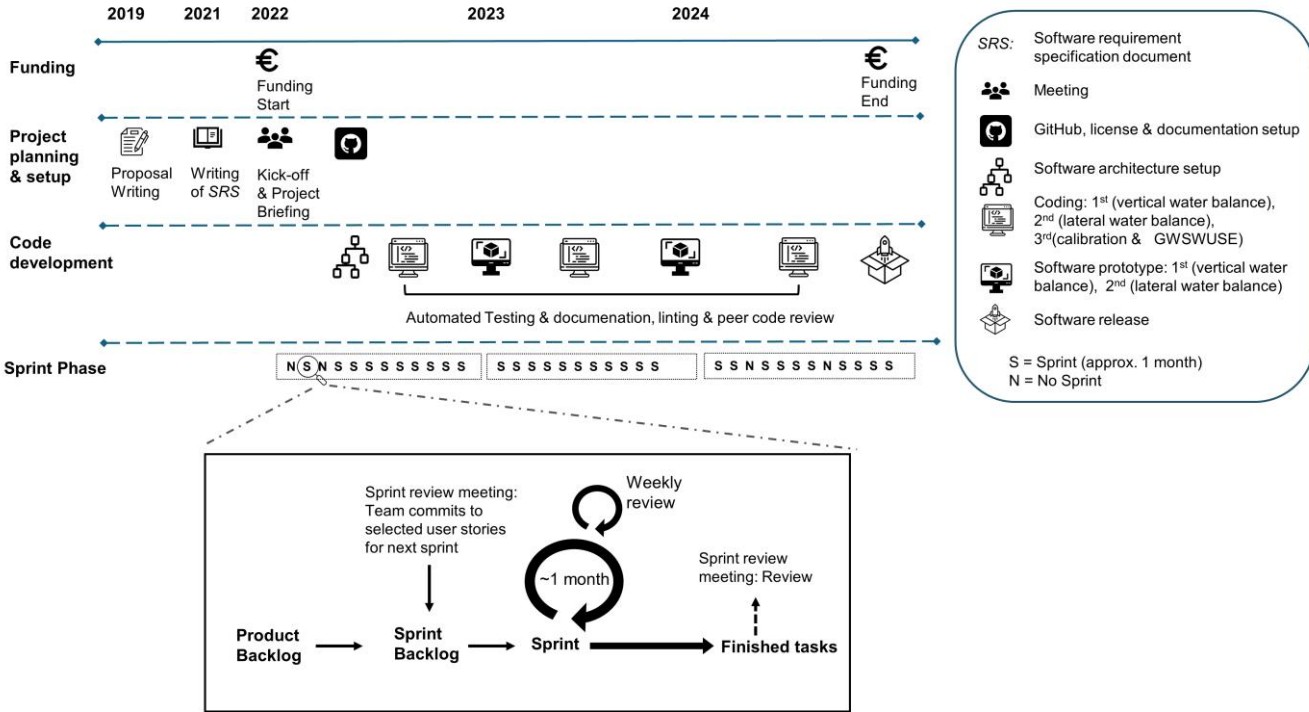

Figure 1: Timeline and agile project management setup for reprogramming of WaterGAP in the ReWaterGAP project.





### 4.1 Project management

#### 4.1.1 Project planning and setup

After the writing and approval of the project proposal, a preliminary meeting of six senior developers was held in November 2021 to draft the software requirement specification document (see software_requirement_WaterGAP.pdf in the Supplement) (Fig. 1), which outlined the technical and functional goals of the project (see section 4.2). This was followed by a kickoff meeting to launch the project officially and a project briefing to align the product owners and development team (see section 4.1.2) on the project's purpose (comprehensively transforming the GHM WaterGAP into sustainable research software) (Fig. 1). As part of the planning and setup phase, a GitHub repository was established for version control, and a software documentation webpage was created.

#### 4.1.2 Agile project management

An agile methodology was adopted for software development, dividing the work into iterative sprints (Hema et al., 2020) (Fig. 1). The project spanned 31 sprints, 27 of which were focused on code development (see section 4.3), with the remainder allocated for reading project materials. Each sprint lasted approximately one month, except for one instance in 2023 when a sprint was extended to two months due to task complexity (Fig. 1). We adopted an agile process inspired by the well-known SCRUM method (Hema et al., 2020), which was tailored to the available resources.

The agile team was comprised of two product owners (the two professors leading the WaterGAP model development, Petra Döll and Martina Flörke), three developers (PhD student Emmanuel Nyenah reprogramming WGHM, Master student Lasse Nissen reprogramming GWSWUSE, and student assistant Leon Mühlenbruch writing the external documentation) and the software development advisor (Robert Reinecke). At the beginning of each sprint, the agile team met to review past progress, with presentation and discussion of completed tasks, and review selected or newly defined user stories (software functionality from the user perspective) which served together with the uncompleted tasks as the basis for the sprint backlog for the next sprint (Fig. 1). User stories were selected from a comprehensive list of user stories and features (product backlog) outlined in the software requirement specification document, which the senior developers wrote before the project started. These user stories were assigned sprint points based on estimated difficulty and time required. A selection of user stories with a combined total of 10 sprint points was then selected to form a sprint backlog.

Progress during each sprint was monitored through weekly meetings between the PhD student and the software development advisor, which provided an opportunity to address challenges encountered during the sprint (e.g., improving runtime of snow module). The agile process allowed the team to maintain steady progress toward the project goals and adapt to new requirements and challenges.



### 4.1.3 Tracking programming effort and progress

The use of agile project management facilitated the tracking of progress on user stories implementation and the corresponding effort (working hours) during code development. While the reprogramming of GWSWUSE was not included in the initial reprogramming scope, it was later included without the setup of user stories. As a result, progress tracking and effort 300 measurement are only limited to the reprogramming of WHGM (see progress_taking.xlsx in the Supplement). 24 major user stories for the WGHM were implemented across 27 sprints and organized into three main phases, the programming of the vertical water balance, the lateral water balance and the calibration routine (Fig. 2). The vertical water balance phase involved implementing key WGHM algorithms such as net radiation, Priestley-Taylor evapotranspiration (PET), and processes related to canopy, snow, and soil. A total of 14 user stories were completed over five sprints, requiring 872 working hours (Fig. 2). 305 The lateral water balance phase focused on the implementation of groundwater, lake, and wetland algorithms, among many others. In total, eight user stories were completed during 18 sprints, requiring 1,800 working hours. A substantial portion of this effort (816 working hours across seven sprints) was dedicated to developing the surface and groundwater abstraction algorithm (see section 5.1.2). The Calibration phase focused on assessing the global water balance and implementing the calibration scheme. This phase was completed within seven sprints, requiring 608 working hours ).


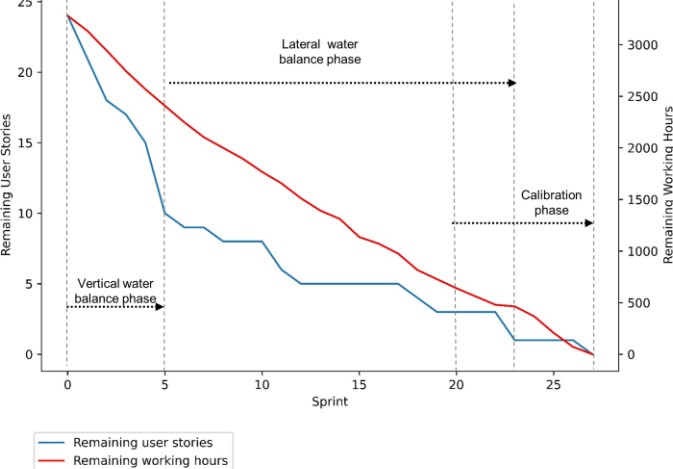

Figure 2: Cumulative plot of remaining user stories and corresponding working hours over 27 sprints (left), and a table of 24 major user stories (right), grouped by their corresponding phases: vertical water balance, lateral water balance, and calibration.



## 4.2 Software requirement specification


The software requirement specification document outlines the intended purpose, features, and functionality of the software, providing guidance for the development process (see software_requirement_WaterGAP.pdf in the Supplement). Key elements of the document are described below.

### 4.2.1 Software architecture and programming language

The selected architectural framework is the Model-View-Controller pattern, which organizes software components into coherent and modular structures with well-defined functionalities (Guaman et al., 2021). In this architecture, the Model component manages the core logic, encompassing hydrological equations and assumptions and managing data for computed results. The Controller component controls the data flow into the Model and facilitates user interactions (e.g., by a configuration file). The View component presents outputs in various formats tailored to user needs, such as saving output in

NetCDF format. Section 5.1 shows a detailed Model–View-Controller diagram of WGHM, illustrating the class, function, and package interactions.

The reprogrammed software is written in Python, chosen for its readability, extensive community support, and a rich ecosystem of libraries (Oliphant, 2007). The Python ecosystem includes packages such as Xarray for handling NetCDF files and NumPy

for efficient array computations, as well as tools that enable parallel computing (Harris et al., 2020; Hoyer and Hamman, 2017; Virtanen et al., 2020).

### 4.2.2 Flexible spatial resolution and restart at prescribed initial state

The new software should allow flexible changes in spatial resolution (30 arc-minutes or 5 arc-minutes). The software should be programmed in such a way that model states, such as storages and parameters, can be saved to disk. A new model run can

then be started from this prescribed initial state. This feature has applications for near-real-time monitoring, ensemble forecasts and data assimilation (e.g., with Parallel Data Assimilation Framework (PDAF))(Müller Schmied et al., 2024).

### 4.2.3 Data formats and user interfaces

The reprogrammed software should have a configuration file in JSON format, while climate forcing and model outputs are stored in NetCDF format. Using the NetCDF format avoids the need for conversion tools, making the software easier to use

and reducing complications. Users interact with the new software through a command-line interface (CLI) and configuration file that can be used to change the model inputs and outputs.





### 4.2.4 User stories

The software requirement specification document also outlines what the software should do from the user's perspective, known as the user story (Curcio et al., 2018; Dimitrijević et al., 2015). The use of user stories ensures that the needs and expectations

of users are met. An example user story is "*As a user, I want the canopy storage and related fluxes to be functionally implemented, based on the model's description paper of WaterGAP2.2d and 2.2e.*" The requirement specification document includes multiple user stories that capture various functional and non-functional requirements. Functional requirements specify what the software should accomplish (e.g., compute canopy algorithm) (Curcio et al., 2018), while non-functional requirements are quality constraints such as correctness, reusability, and maintainability that the software must meet  (Curcio et al., 2018;

Muhammad et al., 2023).

### 4.3 Code development.

### 4.3.1 Best practices for code development

*Meaningful and consistent variable names.* A critical aspect of the code development process was the establishment of

consistent variable names. This was achieved through collaboration between developers, product owners, and the software development advisor. The resulting variable names are descriptive, logical, and uniform across the entire code, significantly enhancing readability and facilitating future maintenance. Examples of variable names can be found here (https://hydrologyfrankfurt.github.io/ReWaterGAP/Glossary/Index.html#glossary)

*Version control and automation.* The project uses Git for version control, with GitHub as the platform for hosting the codebase. The source code of the reprogrammed software is open source and licensed under LGPL v3.0. Throughout the development process, bugs identified in codes through peer code reviews were tracked using GitHub's issue tracking tool, which facilitates transparent tracking of progress and resolution. GitHub Actions were implemented to automate documentation updates, linting, and testing processes. This automation ensures that documentation remains current and maintains code quality through

automated quality checks and testing prior to commits, significantly reducing the likelihood of errors in the main codebase.

*In-code documentation.* Both inline comments and docstrings were used for in-code documentation. Algorithms within the source code were carefully documented with inline comments, explaining the steps and assumptions underlying key processes. Docstrings, on the other hand, are structured comments at the function, class, and module levels that provide a general

description of the code's purpose, parameters, and return values (Wiggins et al., 2023).  They offer a quick reference for developers interacting with the code and can be extracted by documentation generation tools like Sphinx to create external documentation. The in-code documentation was done with the aim to make the code comprehensible for new developers and easy to maintain over time.



*External documentation.* In addition to in-code documentation, a comprehensive web-based documentation (https://hydrologyfrankfurt.github.io/ReWaterGAP/) was generated using GitHub Actions, which facilitates the creation of automated workflows, and the Sphinx library (https://www.sphinx-doc.org/en/master/), which is designed for creating well-structured documentation. The automated workflow is set up through a YAML script (https://github.com/HydrologyFrankfurt/ReWaterGAP/blob/main/.github/workflows/docs_pages.yml) that utilizes Sphinx

library to automate the documentation process. A key feature of Sphinx is its ability to extract docstrings from classes and functions, enabling developers to expand on these docstrings with additional content such as figures, equations, underlying assumptions, and explanations of solution methods (both analytical and numerical). The Sphinx-generated documentation for the reprogrammed software includes an installation guide, tutorials, a user guide, a reference guide, a contributor's guide, frequently asked questions (FAQs), and a glossary. This web-based documentation is automatically updated in tandem with

source code modifications, ensuring that the documentation consistently reflects the current state of the software. Automation for external documentation is currently only available for WGHM and that of GWSWUSE will be added later.

*Logging.* A logging system was added to the project to help with error handling and debugging. Errors and warnings are recorded in a log file, making it easier to troubleshoot issues. The logging system can be adjusted to different user needs,

controlling the amount of detail saved. Additionally, the reprogrammed software can be run in debug mode, showing detailed information on the screen and in the log file. This helps users easily spot and fix problems during use.

*Specialized library.* Given the computationally intensive nature of WGHM software, minimizing its run time was crucial. To achieve this, the Numba library (https://numba.readthedocs.io/en/stable/index.html) was used. Numba is a Just-in-Time (JIT)

compiler for Python that can significantly enhance performance by converting Python functions into machine code at runtime (Lam et al., 2015). Numba operates using function decorators defined as wrapper functions that inform Numba regarding the specific functions that should be compiled into machine code (Lam et al., 2015).

*Containerized solution for WGHM.* A Dockerfile following best practices for writing Dockerfile (Nüst et al., 2020) is available

to create a containerized environment for running WGHM (https://github.com/HydrologyFrankfurt/ReWaterGAP/blob/main/Dockerfile). This Dockerfile sets up a Python environment with the required packages and clones the source code of the reprogrammed software during the build of a Docker image (executable file). Once the image is built, it can be run (the running instance of the image is known as a container (Nüst et al., 2020)). A simple tutorial on running the WGHM Docker container can be found here

(https://hydrologyfrankfurt.github.io/ReWaterGAP/user_guide/tutorials/tutorial_running_with_docker.html#tutorial-docker)





### 4.3.2 Quality assurance

*Automated unit testing.* Unit testing is a software development process in which individual pieces of code (units) are tested to ensure they function as expected (Pajankar, 2022). If the code is changed at a later stage, e.g., as new features are added, the automated execution of the test reduces the likelihood that something is "broken" without the developers noticing it.

Furthermore, tests can help to develop new software features according to a specification, also called test-driven development, in which tests are written before implementing the software component (George and Williams, 2004). The code to test a function is called a unit test or test case, while a collection of test cases forms a test suite. Unit testing can be conducted for the reprogrammed WGHM and GWSWUSE components using the Python unittest framework (Pajankar, 2022; Python, 2025). An example of unit tests written for the canopy storage module

(https://github.com/HydrologyFrankfurt/ReWaterGAP/blob/main/test/test_canopy.py) contains

1. a setup function (lines 25-44) that generates randomly plausible input data for testing and stores plausible minimum and maximum daily benchmark values (Müller Schmied et al., 2021),

2. a first unit test (Lines 47–76) which runs the canopy storage module for one day and compares this result against the benchmark, and

3. a second unit test (79-120) to verify whether the canopy storage module raises an error message when it encounters negative precipitation values in a grid cell.

The complete suite of tests for the reprogrammed WGHM software is publicly available (https://github.com/HydrologyFrankfurt/ReWaterGAP/tree/main/test) and that of the reprogrammed GWSWUSE is available

via (https://github.com/HydrologyFrankfurt/ReGWSWUSE/tree/main/test). To automate the execution of test suites, a GitHub Actions workflow has been created using a YAML script. An example script is found here (https://github.com/HydrologyFrankfurt/ReWaterGAP/blob/main/.github/workflows/unit_test.yaml). Automation for testing is currently only available for WGHM, while automated testing of GWSWUSE will be added later. To ensure the testing of model functionality, new tests must be added to the existing test suites whenever new process algorithms are introduced.

Expanding on existing tests, for example, to test additional edge cases, is the first task we recommend when onboarding new developers into an existing project.

*Peer code review.* To enhance code quality, three hackathon-style peer code review sessions were organized. During these events, eight WaterGAP developers examined the WGHM codebase, executed the software, and actively sought out bugs.

These sessions also evaluated the clarity of external documentation, ensuring that it was comprehensible and user-friendly. In addition, several weekly meetings involving developers and the software development advisor were dedicated to code reviews.



*Linting*. To maintain consistency and readability across the code, the reprogrammed software code was checked against PEP-8 conventions, which define the style guidelines for Python code (https://peps.python.org/pep-0008/). The Python library Pylint

was employed to assess the code for potential bugs and deviations from these conventions. Linting is also automated and an example script be found here (https://github.com/HydrologyFrankfurt/ReWaterGAP/blob/main/.github/workflows/lint.yaml). Automation for linting for is currently only available for WGHM.

## 5 Architecture and new features of the reprogrammed software

### 5.1 WGHM

#### 5.1.1 Architecture

In Figure 3, the Model-View-Controller (MVC) architectural pattern  for the reprogrammed WGHM software is shown in its low-level implementation. (Gamma et al., 1994; Guaman et al., 2021) The run_watergap module coordinates the entire model workflow. It manages process initialization and daily time-stepping to perform computations.   The Controller package is responsible for handling configuration-related tasks. The *Config_handler()* function processes information from configuration

files, which includes paths to input data such as climate forcing and static datasets, as well as runtime settings like the simulation period and the type of run (e.g., naturalized or anthropogenic). This information is passed into classes such as *ClimateForcing* and *StaticData*, which read the specified input files and prepare the data for use in the model. For instance, the *ClimateForcing* class accesses and processes precipitation and temperature data, while *StaticData* processes data for land cover and other static variables.


The Model package implements all hydrological processes organized into vertical and lateral water balance components. In the reprogrammed WGHM, each storage compartment is designed as a separate Python module. The *VerticalWaterBalance* class coordinates hydrological processes such as the calculation of net radiation and potential evapotranspiration (PET) using the Priestley-Taylor algorithm (although other PET schemes can be easily incorporated), canopy, snow, and the soil water

balance. The class uses its *calculate()* function to manage these computations and obtain the resulting storages and fluxes through the *get_storages_and_fluxes()* function. Also, the *LateralWaterBalance* class addresses horizontal water movements via *calculate()* function which further calls the *river_routing()* function. This involves the simulation of storage compartments like groundwater, lakes and wetlands, reservoir-regulated water bodies, and rivers. It similarly retrieves all associated storages and fluxes through its *get_storages_and_fluxes()* functions.

Model parameters in NetCDF format are also processed here. The NetCDF format not only facilitates easy visualization of parameter distribution but also enables convenient parameter modification using libraries like Xarray (Hoyer and Hamman, 2017).



The View package processes the outputs generated by the Model. It extracts storages and fluxes through the
*get_storages_and_fluxes()* functions of the *VerticalWaterBalance* and *LateralWaterBalance* classes. Outputs are then converted to base units and saved as NetCDF files. NetCDFs are enriched with metadata, which comply with ISIMIP conventions (https://www.isimip.org/protocol/preparing-simulation-files/).

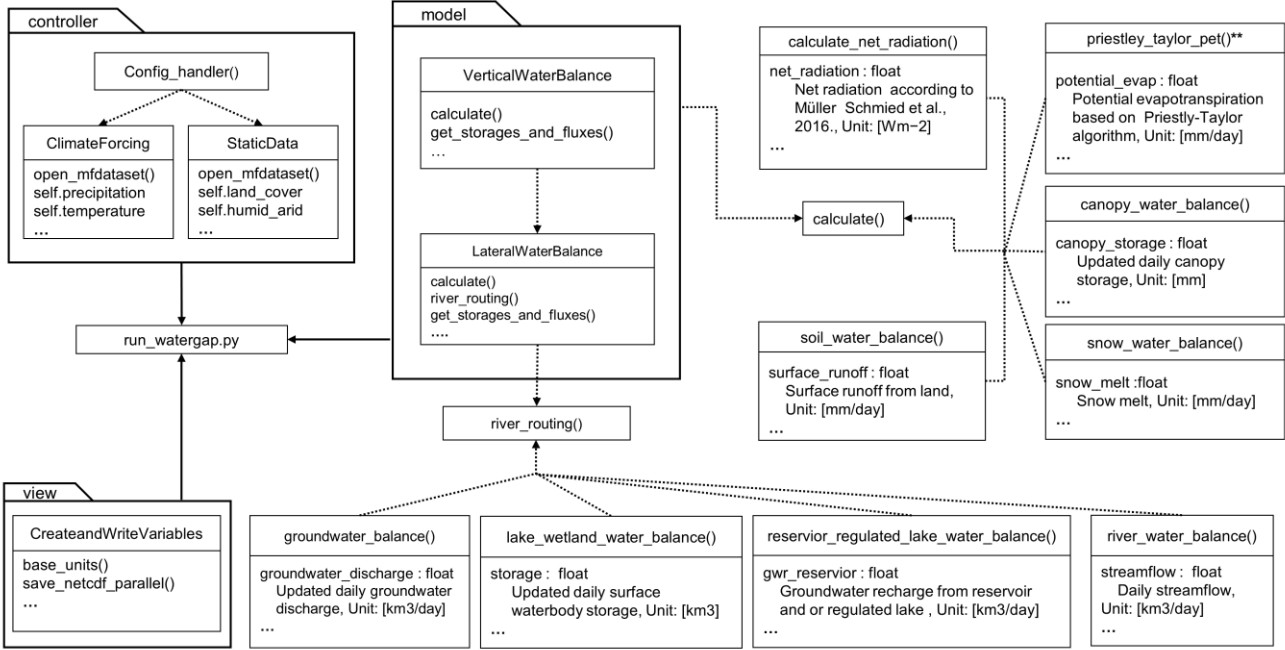

Figure 3: Model-View–Controller (MVC) architectural pattern of the reprogrammed WGHM software at the package, class, and function levels. The Controller package manages the configuration and input data (e.g., climate time series and static data), the Model package contains core hydrological processes, and the View package handles the saving and presentation of model outputs in NetCDF format. Classes are represented with capitalized names, and functions are denoted by lower-case names ending with parentheses.


### 5.1.2 New features

Reprogramming enabled the revision of the algorithm governing surface water demand satisfaction and its impact on return flows to groundwater. The legacy code lacked sufficient in-code and external documentation to enable code comprehension. After discussing the underlying conceptual model with the product owners, we developed an improved consistent algorithm.





For more details about the new abstraction algorithm, readers can refer to its documentation (https://hydrologyfrankfurt.github.io/ReWaterGAP/model_processes/lateral_water_balance/net_abstractions.html).

## 5.2 GWSWUSE

### 5.2.1 Architecture

The reprogrammed GWSWSUE component follows a similar MVC architecture (see Fig. S1 in the Supplement). The source
code is available on GitHub (https://github.com/HydrologyFrankfurt/ReGWSWUSE.git).

### 5.2.2 New features

Reprogramming GWSWUSE software included the modification of model equations to enable the calculation and write-out of potential sectoral net abstractions from groundwater and surface water, and the sectoral return flows to groundwater and surface water while the legacy GWSWUSE code only provided total net abstractions from groundwater and surface water.
Moreover, additional optional model settings were added in the reprogrammed GWSWUSE that enable an improved and updated modelling of irrigation water use, including both water abstractions and consumptive use (i.e. abstracted water that evapotranspirates during use (Müller Schmied et al., 2021).

Most importantly, the reprogrammed GWSWUSE can now optionally handle the input of a new variant of the irrigation water
use model GIM (Müller Schmied et al., 2021) that uses an updated dataset of the time series of area equipped for irrigation (AEI) in each grid cell to compute the consumptive irrigation water use on the AEI. It then implements recent information on country values of area actually irrigated (AAI) and AEI available from AQUASTAT (https://www.fao.org/aquastat/) for the time period 1964-2020 to compute consumptive irrigation water use on the AAI since 1901. While the old gridded AEI dataset covered the period from 1900 to 2005 (Siebert et al., 2015), the new gridded data incorporates new data for 2000 to 2015
(Mehta et al., 2024). Consumptive irrigation water use on AAI in the period 1901-2015 is computed by multiplying the gridded output of GIM by the country-specific ratio of AAI-to-AEI, while results for the period 2016-2020 are computed in reprogrammed GWSWUSE by multiplying the AAI-to-AEI ratio for 2015 by the ratio of AAI in the specific year to the AAI in 2015. Irrigation after 2020 is handled in the new GWSWUSE like 2020. (Müller Schmied et al., 2021)

Another newly included option is an alternative computation of irrigation water abstractions from groundwater. Instead of a
globally valid water use efficiency of 0.7, the user can select that the water use efficiency for irrigation with groundwater is not less than the country-specific water use efficiency for irrigation with surface water.

For more details on these new features and the overall functionality of the new GWSWUSE software, please refer to the external documentation (https://hydrologyfrankfurt.github.io/ReWaterGAP/model_processes/gwswuse/index.html#gwswuse)
and Fig. S2 in the Supplement



## 6 Evaluation against sustainability criteria and FAIR4RS principles

The reprogrammed WGHM and GWSWUSE software demonstrate significant improvements in software engineering practices and source code quality compared to the legacy software. They include a more comprehensive external
documentation, which was absent in the legacy software (Table 1). While the legacy and reprogrammed software use GitHub for version control, the latter is publicly accessible and includes automation (currently for WGHM) for documentation, testing, and linting (Table 1). The reprogrammed software provides containerization (currently for WGHM), which was unavailable for the legacy software (Table 1). Regarding active development over the past two years, both the reprogrammed and legacy WGHM software have seen ongoing development, with three active developers working on the reprogrammed version and
four developers maintaining the legacy version. Research projects continue to rely on the legacy WGHM codebase; thus, development activities are expected to continue until a smooth transition to the reprogrammed software is achieved. In contrast, no active development has occurred for the legacy GWSWUSE model in the past two years.

Regarding source code quality, the reprogrammed WGHM includes a publicly available automated testing suite, which ensures
components of the software function as intended (Table 1). The reprogrammed software programs comply with Python PEP-8 coding standards, with a Pylint score of 9.40 (out of 10) for WGHM and 9.65 for GWSWUSE. In contrast, the legacy software contains several warnings, typos, and errors when evaluated against C/C++ Core guidelines leading to potential issues like poor code readability and difficulty in maintenance (Table 1). Comment density has improved for WGHM from 21% in the legacy software to 47% in the reprogrammed software, improving readability and enabling easier maintenance (Table 1,
Fig. S3a in Supplement). However, the comment density in GWSWUSE decreased from 50% to 26%, even though it was sufficient for code comprehension. This decline is partly because developers in the legacy version of GWSWUSE recorded file history in the headers, which increased the number of comment lines. Based on the modularity metric, the legacy software programs include several files that exceed the recommended range of 10–1,000 TLOC per file (see Fig. S3b in Supplement). The reprogrammed software have a modular structure, keeping TLOC per file within the recommended limits (see Fig. S3b in
Supplement).






Table 1: Sustainability indicators for the legacy and reprogrammed WGHM Software

| No. | Indicator | Legacy WGHM & GWSWUSE | Reprogrammed WGHM & GWSWUSE |
|-----|-----------|-----------------------|------------------------------|
| Best practice in software engineering | | | |
| 1 | External documentation | No | Yes |
| 2 | Version control and automation | Yes, GitHub (private), no automations available | Yes, GitHub (public), automation for documentation, testing and linting (currently for WGHM) |
| 3 | Open-source license | LGPLv3 | LGPLv3 |
| 4 | Number of active developers | WGHM = 4 , GWSWUSE = 0 | WGHM = 3,  GWSWUSE =1 |
| 5 | Containerization | No | Yes (currently for WHGM) |
| Source code quality | | | |
| 6 | Public availability of an automated testing suite | No | Yes (currently only for WGHM) |
| 7 | Compliance with coding standard | Several code violations: WGHM (~280 warnings, ~3600 typos, 140 errors),  GWSWUSE (~70 warnings, ~860 typos, 5 errors) | Yes, WGHM Pylint score = 9.40/10 , GWSWUSE Pylint score = 9.65/10 |
| 8 | Comment density | WGHM = 21%, GWSWUSE = 50% | WGHM = 47%, GWSWUSE = 26% |
| 9 | Modularity | No | Yes |


The reprogrammed software aligns with the eleven main FAIR4RS principles. It has a versioned DOI from Zenodo (FAIR4RS principle F1, Barker et al., 2022), along with rich metadata such as web-based documentation (F2) that includes the DOI (F3). Metadata is searchable and indexable (e.g., via a search engine) (F4). The software can be downloaded from both the GitHub repository and Zenodo (A1), and the metadata will remain accessible even if the software becomes unavailable on Zenodo 555 (A2). The software uses data types (NetCDF) for input, output, and data exchange that are widely used in the global hydrological and impact model community (I1). The software includes qualified references to other objects (e.g., climate forcing data) (I2). It is published under a Lesser General Public License v3.0 (R1). The code also includes qualified references to other software, such as various Python libraries (e.g., Xarray, Numpy, Numba) (R2), and the software meets domain-relevant community standards (e.g., variable naming convention from ISIMIP) (R3).






## 7 Differences between the outputs of the reprogrammed and legacy software

Globally aggregated water balance components (km$^3$ yr$^{-1}$) are shown for the reprogrammed version and the legacy code in Table 2. As expected, the output of both WGHM versions is very similar. The most notable changes are in the actual net abstraction from surface water and groundwater, which can be attributed to the implementation of a consistent water abstraction

algorithm in WGHM and small variations in the outputs of the reprogrammed GWSWUSE compared to the legacy version. Additionally, the use of a new minimization algorithm for parameter calibration, the "Powell method" from SciPy (https://docs.scipy.org/doc/scipy-1.15.0/reference/optimize.minimize-powell.html), has contributed to the overall variations in the water balance components (see Fig. S4 in the Supplement for the variations in calibrated parameters contributing to variation in water balance components).


Table 2: Global-scale (excluding Antarctica and Greenland) water balance components (km$^3$ yr$^{-1}$) for the reprogrammed and legacy WaterGAP global hydrological models, driven by the climate forcing data from gswp3-w5e5. Long-term average volume balance error is calculated as the difference in component 1 and the sum of components 2, 3, and 8. Values within

parentheses are for the legacy WGHM.

| No. | Component | 1961-1990 | 1971–2000 | 1981–2010 | 1991–2019 | 2001–2019 |
|---|---|---|---|---|---|---|
| 1 | Precipitation | 110637 (110637) | 111279 (111279) | 111350 (111350) | 111574 (111574) | 111655 (111655) |
| 2 | Actual evapotranspiration | 71427 (71325) | 71861 (71755) | 71926 (71816) | 72113 (71998) | 72179 (72063) |
| 3 | Streamflow into oceans | 39199 (39295) | 39432 (39530) | 39484 (39584) | 39563 (39666) | 39593 (39697) |
| 4 | Inflow into inland sinks | 774 (776) | 793 (794) | 794 (795) | 840 (841) | 845 (846) |
| 5 | Actual consumptive water use | 909 (904) | 1055 (1049) | 1203 (1195) | 1316 (1307) | 1379 (1369) |
| 6 | Actual net abstraction from surface water | 1000 (1036) | 1140 (1186) | 1282 (1338) | 1386 (1448) | 1434 (1501) |
| 7 | Actual net abstraction from groundwater | -91 (−132) | -85 (−137) | -79 (−143) | -70 (−141) | -55 (−132) |
| 8 | Change in total water storage | 11 (17) | -14 (−6) | -59 (−49) | -102 (−91) | -117 (−105) |
| 9 | Long-term average volume balance error | 0.11 (−0.46) | 0.11 (−0.34) | 0.09 (−0.20) | 0.09 (−0.08) | 0.09 (−0.07) |



The differences between the grid cell values of renewable water resources between the reprogrammed software (Fig. 4a) and the legacy software are small. For 97.6% of the global land area, the difference remains within ±10% (Fig. 4b). More

specifically, 70.5 of the global land area has renewable water resources that differ within ±1 % while 27.1% of the global land area falls within the range of ±1% and ±10% (Fig. 4b). Only 0.1% of the global land area shows relative difference exceeding ±100%. The differences between the legacy and reprogrammed versions for renewable water resources are only due to variations in calibration parameters


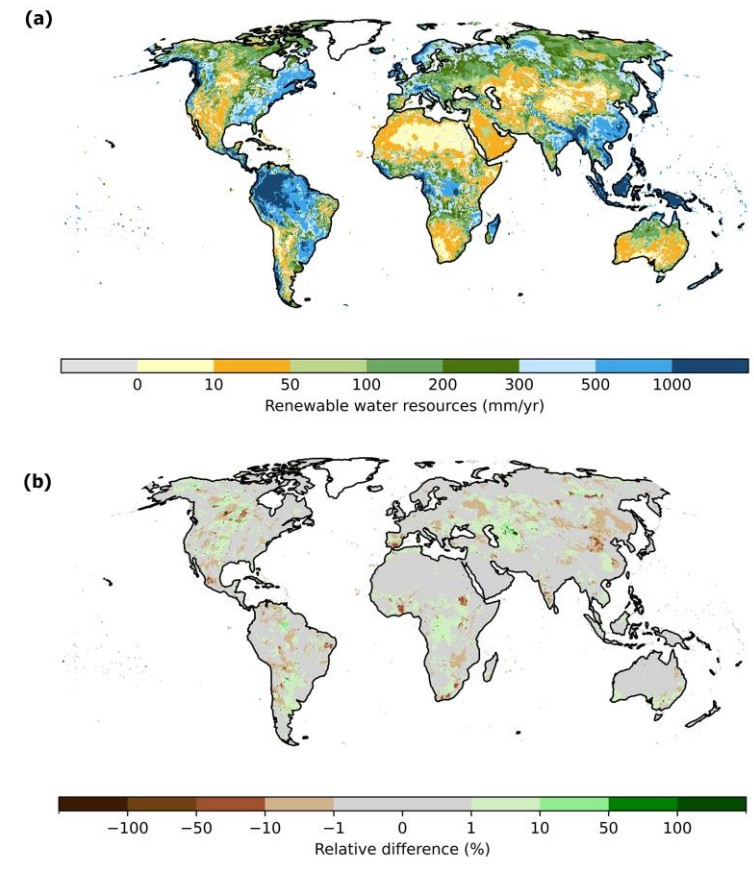

Figure 4: Total renewable water resources [mm yr$^{-1}$] for the period 1981–2010 computed by the reprogrammed WaterGAP model (a), percent differences of computed total renewable water resources between the reprogrammed and legacy WaterGAP model for the period 1981–2010. Positive values in b) indicate that the legacy WGHM estimates higher renewable water resources than the reprogrammed WGHM.






## 8 Users perceptions of the reprogrammed WaterGAP software

The online survey aimed to evaluate the readability, comprehensibility, modifiability, and documentation quality of a code snippet implementing the Priestley-Taylor potential evapotranspiration algorithm in the reprogrammed software. The 64 survey participants who completed the survey represented a diverse group, with the majority being PhD students (38%), 595 scientific staff (20%), or Postdocs (14%) (see Fig. S5a in the Supplement). On average, participants had approximately 14 years of programming experience, with individual experience ranging from 1 to 50 years (see Fig. S5b in the Supplement).

The survey results demonstrate a high level of code readability, with about 72% of the participants correctly identifying the Priestley-Taylor algorithm (see Fig. S6a in the Supplement). Approximately 72% of the participants agreed (with 50% strongly 600 agreeing) that the provided external documentation clearly explained the code (Fig 5). Furthermore, about 58% of participants understood of the algorithm's purpose after reading the external documentation, and about 64% confirmed that the documentation was not difficult to comprehend (Fig 5). The survey also examined the ease of code modification by testing participants' confidence in implementing a change to the algorithm. When asked about modifying the code with a new atmospheric constant, approximately 62% of participants expressed some level of confidence (ranging from slightly to very 605 confident) in their ability to do so (see Fig. S6b in the Supplement). In contrast, 27% did not respond, 5% indicated having no confidence, and 6% stated they had no idea how to proceed (see Fig. S6b in the Supplement). Additionally, about 66% correctly identified the specific line of code that would require modification (see Fig. S6c in the Supplement). Meanwhile, about 26% did not respond, and 8% selected the wrong line of code.

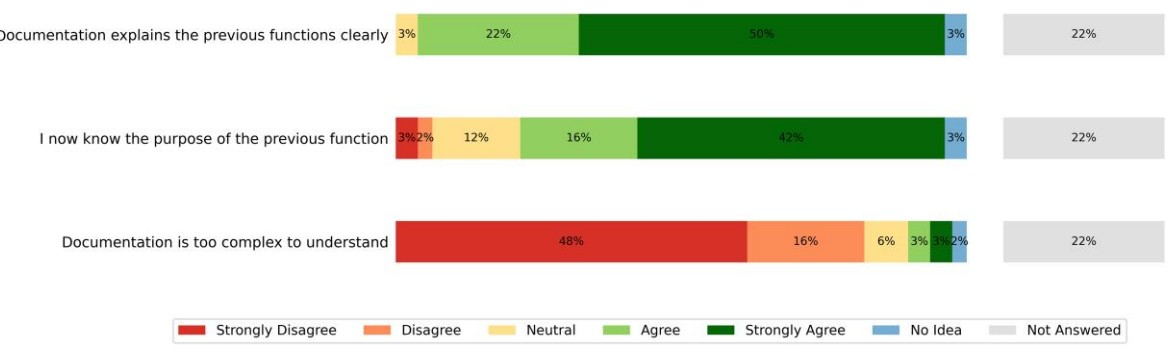


Figure 5: Survey results from 64 participants regarding external documentation readability and comprehension of the Priestley-Taylor potential evapotranspiration (PET) code snippet.



## 9 Discussion

Considering sustainable research software indicators and the FAIR4RS principles in the reprogramming of the legacy code has enhanced the software quality, extensibility, reproducibility, and long-term sustainability of WaterGAP. Unfortunately, the transition from C/C++ in the legacy software to Python, an interpreted language, has approximately doubled the WGHM runtime. This is to be expected as numerical computations in Python can be 3-10 times slower compared to C/C++ (Cai et al., 2005). The average runtime for a standard run on an AMD EPYC 7543 processor with 3.7 GHz is about 7-8 minutes per 620 simulated year for the reprogrammed software, compared to 3-4 minutes for the legacy software. Considering that this may lead to critical run time-related constrains for model calibration and ensemble methods, e.g., used for sensitivity analysis and ensemble forecasts – is the choice of Python justifiable?

To reach this runtime, we already utilized the optimization library Numba, which compiles parts of the Python code. Python 625 is generally slower in terms of runtime performance compared to C/C++ since it uses interpretation instead of compilation (Cai et al., 2005). Compiled code is translated into machine code by a compiler before execution, resulting in a standalone executable file that can be run directly by the processor. On the other hand, interpreted code is executed line by line by an interpreter during runtime, meaning the code must be interpreted every time it is run. Compiled code generally executes faster but often requires a separate compilation step and may be less portable. In contrast, interpreted code is typically more portable 630 but executes more slowly. The pure Python implementation of the GHM HydroPy model is three times slower than the version with a routing scheme written in Fortran (Stacke and Hagemann, 2021).

However, Python generally produces more readable, less error-prone, and more maintainable code than C++, primarily due to its simpler syntax, dynamic typing, automatic memory management, and higher-level abstractions (Balreira et al., 2023; 635 Johnson, 2025; Prechelt, 2000). These features reduce the likelihood of errors and allow developers to express complex ideas more concisely. Python's extensive standard library and ecosystem further enhance maintainability by reducing the need for custom code. In contrast, C++'s more complex syntax and manual memory management can lead to more errors and harder-to-maintain code. Most scientists lack the necessary skills to produce high-quality C++ code and are unlikely to follow any best practices (Reinecke et al., 2022). We believe that the benefits of Python regarding of code quality outweigh the run time 640 increase. The switch from C/C++ to Python makes it easier for scientists, particularly those with restricted programming experience, to understand, modify, extend and maintain a complex model. Slow code can always be made fast with better hardware, but hardware cannot fix bad code and unsustainable software.

The agile development process and the use of user stories were essential in our reprogramming effort, enabling iterative 645 improvements through continuous feedback. User stories helped ensure that the software features matched the scientific requirements of WaterGAP. However, estimating the time needed to complete user stories was difficult, and coordinating an





agile process with only a few developers in an academic setting was somewhat challenging. Despite this, we recommend this approach for other reprogramming projects as it supports user-focused development and helps the team stay updated on progress.


Our user survey has several limitations and potential biases. The survey was distributed to participants at the European Geosciences Union (EGU) 2024 conference, which introduced self-selection bias. Respondents were likely more interested in software sustainability topics, potentially skewing the results toward a more engaged subset of researchers. More importantly, we assessed understanding and perceptions based on a code snippet rather than the full source code, which does not provide a

comprehensive evaluation of the software. Due to time constraints, we did not conduct a practical evaluation of reproducibility, with guiding participants through executing the reprogrammed software with a tutorial.

## 10 Conclusion

This study details how the legacy software of the state-of-the-art global hydrological model WaterGAP was reprogrammed to obtain a sustainable research software that can be efficiently applied and enhanced within the original developer group and a

broader research community. The reprogrammed software (from the so-called ReWaterGAP project) has undergone extensive quality control, which has enhanced its reliability. The new modular structure and use of the Python programming language have greatly improved readability, modifiability, and extensibility. The software quality, comprehensive documentation, containerization, and use of standardized input and output formats make the software more accessible to users with varying levels of expertise. The open-source nature of the reprogrammed software allows for easier comparison of algorithms,

consistency checks, and error detection, ultimately contributing to the advancement of hydrological sciences.

With the reprogrammed WaterGAP software, interested researchers can now be taught at conferences or summer schools about how to apply and improve the software, thus expanding the WaterGAP community and advancing global hydrological modeling. Moreover, the reprogrammed software, with its improved modularity, can serve for teaching Bachelor and Master

students to write and modify algorithm or include new input data, gaining new insights into the global hydrological cycle.

**Code and data availability**

The reprogrammed WGHM model source code as well as Docker file can be found at https://github.com/HydrologyFrankfurt/ReWaterGAP. An archived release of the reprogrammed WGHM is also made

available on Zenodo (https://doi.org/10.5281/zenodo.14988011) (Nyenah et al., 2025a). The source code of the new GWSWUSE is available on GitHub (https://github.com/HydrologyFrankfurt/ReGWSWUSE.git) as well as Zenodo (https://doi.org/10.5281/zenodo.14988011) (Nyenah et al., 2025a). External documentation for both source code can be



accessed via (https://hydrologyfrankfurt.github.io/ReWaterGAP/). The Python scripts utilized for analysis is available on Zenodo (https://doi.org/10.5281/zenodo.14988257) (Nyenah et al., 2025b).


**Author contributions**

EN, PD and RR designed the study. EN performed the analysis and wrote the paper with significant contributions from PD, MF, LM, LN and RR. RR and PD supervised EN.

**Acknowledgments**

The study was supported by a grant of the Deutsche Forschungsgemeinschaft (DFG) (grant no. 443183317). We  thank Linda Söller and Laura Müller for their valuable advice for the development of the user survey

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
