# Peer review of "The Process and Value of Reprogramming a Legacy Global Hydrological Model"

_EGUsphere, 2025_

## Referee Comment (RC1)

**Review of The Process and Value of Reprogramming a Legacy Global Hydrological Model**

Article by Nyenah et. al. Review by Rolf Hut

The authors present their work on reprogramming the WaterGAP model from its legacy code in C/C++ into python. The main purpose of this activity is to enhance the reproducibility of science done with WaterGAP, to make the science more transparent (FAIR) and overall reduce the effort required to maintain such a large code-base.

This is a worthwhile effort that will greatly help the (hydrological) scientific community in general and the WaterGAP user base in particular. The python version of WaterGAP has potential for bigger uptake, easier collaboration and is generally a better piece of research software than the legacy C/C++ version.

I do, however, struggle with this publication and its place in the academic literature. I find it hard both to judge what the intention of the authors is with the publication and if they success in that.

*Sidenote: reporting on the progress of software projects within academia is a struggle. The classic "report on results of work done so others can build on it"-structure of academic articles doesn't fit on reporting on developing new software, because the software in and of itself is not a scientific result. I would argue, however, that in the current age where academic credit is almost solely awarded based on "publications" a form of reporting on important software projects is needed. Both for informing the academic community on the new software (availability) and for rewarding / acknowledging those working on building that software. We had exactly the same problem when writing our eWaterCycle papers, where the first eWaterCycle paper never made it past peer review because it lacked "scientific results". In the second paper we focused on providing use cases to illustrate the platform and did manage to publish the work. I think this illustrates that GMD as a journal is accepting more and more software-like contributions.*

Below I will list the different purposes I identified in the manuscript and provide feedback and tips for each different purpose to optimize the paper towards that purpose. I leave it to the combined team of authors and the editors of GMD to decide on which purpose they want to prioritize in the manuscript (or maybe even split in different manuscripts, for different audiences, using different platforms?)

**Announcing reprogrammed version of WaterGAP for potential users**

The new WaterGAP seems to me like an amazing tool for hydrologists to work with. Sections 2.1 (Model description), 5 (architecture), 6 (eval against sustainability criteria) , 7 (Difference between output of C/C++ and Python versions) are essential for communicating this. For this focus I strongly suggest to add a few case studies that demonstrate the capabilities and user friendly-ness of the new WaterGAP.

**Reporting on the process of reprogramming a legacy model**

For those that contemplate reprogramming a legacy model, lessons learned from the transition from C/C++ version of WaterGAP to Python are very valuable. Section 3.1 on software evaluation against (FAIR) criteria and section 4 on the reprogramming process are very valuable here. I would add a paragraph on "lessons learned" that give pointers for others that set out to undertake a similar effort.

**Reporting on user experience with the new WaterGAP codebase**

The overview gathered from the survey conducted at EGU (section 3.3 and 8) gives some preliminary info on the perception of (potential) users of the new WaterGAP code towards its quality. This is in principle a valuable addition to the literature, but the width and execution of the survey is slim for a stand-alone publication. The selective response and low number of respondents make generalizing claims from this survey hard. I would strongly advice to present the results in terms of absolute numbers instead of percentages, so "10 people thought it was easy" versus "15% of people thought it was easy". For a full report on how outside users experience the new WaterGAP I suggest additional work, including for example a focus group session where users working with the new WaterGAP are observed.

**Minor remarks**

Independent of the direction chosen I have a few smaller remarks on the current version of the text:
- Line 61, add a citation to Wilkinson 2016 for FAIR
- Line 65: the unsuspecting reader might conclude that "to improve … … were reprogrammed" was done as part of the aforementioned eWatercycle project
- Line 181 and continuing: I would avoid as much as possible using URLs as citations and use bibtex citations to websites (preferably with a "last accessed" field)
- Section 4.1.2. I always learned that agile is a project management tool for when you have a fixed amount of hours, or people, to work on something, but the end goal is not fixed, just "create as much value as quickly as possible" (ideal for start-up culture). This is ideal when project goals are allowed to fluctuate during the project. In most scientific large projects the end goal is quite fixed (reprogram this model into python). Therefore, I would invite the authors to highlight why they settled on Agile as a project management practice for this project (there might well be very good reasons!)
- Line 288: "senior developers" are introduced as a role, but not previously explained.
- Line 291: same for "software development advisor"

**Concluding**

I really like the new python version of WaterGAP. (I like it so much that I would invite the authors to work together to add support for WaterGAP in eWaterCycle to make it even more accessible to hydrologists!). I hope the above suggestions will help in choosing a focus direction for the manuscript and emphasizing those parts throughout. I am happy to review an updated version of this manuscript.

---

## Author Response (AR1)

Dear Reviewers,

We sincerely appreciate your prompt and insightful review of our manuscript. Your valuable comments and suggestions have significantly improved the quality of our manuscript. Below, we address each of your points in detail and outline the corresponding changes made to the manuscript. For clarity, your comments are highlighted in blue, our responses are in black, and any newly added text appears in *italics*. All sections and line numbers refer to the revised version of the manuscript.

To improve our manuscript, we have
- revised key aspects of the Introduction and Methods sections (Sections 1 and 3, respectively). In particular, we have reformulated the manuscript's objectives.
- simplified the structure of the manuscript, moving the user survey to the supplement.
- enhanced the sections describing the programming process (Section 4) to avoid repetition with the Methods section.
- included a "Lessons learned" section (section 8) to benefit others undertaking similar efforts.

**Reply to Reviewer 1:  Rolf Hut**

The authors present their work on reprogramming the WaterGAP model from its legacy code in C/C++ into python. The main purpose of this activity is to enhance the reproducibility of science done with WaterGAP, to make the science more transparent (FAIR) and overall reduce the effort required to maintain such a large code-base.

This is a worthwhile effort that will greatly help the (hydrological) scientific community in general and the WaterGAP user base in particular. The python version of WaterGAP has potential for bigger uptake, easier collaboration and is generally a better piece of research software than the legacy C/C++ version.

Thank you for highlighting the importance and quality of our paper.

I do, however, struggle with this publication and its place in the academic literature. I find it hard both to judge what the intention of the authors is with the publication and if they success in that.

Sidenote: reporting on the progress of software projects within academia is a struggle. The classic "report on results of work done so others can build on it"-structure of academic articles doesn't fit on reporting on developing new software, because the software in and of itself is not a scientific result. I would argue, however, that in the current age where academic credit is almost solely awarded based on "publications" a form of reporting on important software projects is needed. Both for informing the academic community on the new software (availability) and for rewarding / acknowledging those working on building that software. We had exactly the same problem when writing our eWaterCycle papers, where the first eWaterCycle paper never made it past peer review because it lacked "scientific results". In the second paper we focused on providing use cases to illustrate the platform and did manage to publish the work. I think this illustrates that GMD as a journal is accepting more and more software-like contributions.

Below I will list the different purposes I identified in the manuscript and provide feedback and tips for each different purpose to optimize the paper towards that purpose. I leave it to the combined team of authors and the editors of GMD to decide on which purpose they want to prioritize in the

manuscript (or maybe even split in different manuscripts, for different audiences, using different platforms?)

**Announcing reprogrammed version of WaterGAP for potential users**

The new WaterGAP seems to me like an amazing tool for hydrologists to work with. Sections 2.1 (Model description), 5 (architecture), 6 (eval against sustainability criteria), 7 (Difference between output of C/C++ and Python versions) are essential for communicating this. For this focus I strongly suggest to add a few case studies that demonstrate the capabilities and user friendly-ness of the new WaterGAP.

**Reporting on the process of reprogramming a legacy model**

For those that contemplate reprogramming a legacy model, lessons learned from the transition from C/C++ version of WaterGAP to Python are very valuable. Section 3.1 on software evaluation against (FAIR) criteria and section 4 on the reprogramming process are very valuable here. I would add a paragraph on "lessons learned" that give pointers for others that set out to undertake a similar effort.

**Reporting on user experience with the new WaterGAP codebase**

The overview gathered from the survey conducted at EGU (section 3.3 and 8) gives some preliminary info on the perception of (potential) users of the new WaterGAP code towards its quality. This is in principle a valuable addition to the literature, but the width and execution of the survey is slim for a stand-alone publication. The selective response and low number of respondents make generalizing claims from this survey hard. I would strongly advice to present the results in terms of absolute numbers instead of percentages, so "10 people thought it was easy" versus "15% of people thought it was easy". For a full report on how outside users experience the new WaterGAP I suggest additional work, including for example a focus group session where users working with the new WaterGAP are observed.

Thank you very much for your thoughtful and constructive feedback. We appreciate your recognition of the value of our work and your insights into the challenges of publishing software-focused contributions in academic literature. Our goal, in fact, was not to announce the reprogrammed version of the WaterGAP but to report on the process of reprogramming a legacy model into a sustainable research software and to scientifically investigate the value of reprogramming scientific software. The main intention of the manuscript is to provide guidance for those who wish to reprogram the legacy code of the scientific model into sustainable software that can be easily maintained and improved, thus improving the reproducibility of the computational research done with this software. The comments provided by you and Reviewer 2, however, highlight that we did not succeed in conveying that in our initial manuscript.

We have revised the manuscript based on your feedback and our own reflections to focus explicitly and strongly on "Reporting on the process of reprogramming a legacy model". As you highlighted, the lessons learned from transitioning the WaterGAP model from C/C++ to Python are indeed valuable for undertaking similar efforts. In response, we have removed the previous discussion section and have included a dedicated "Lessons learned" section. This new section, along with sections 3 on software sustainability and the FAIR principles, and section 4 on the reprogramming process, significantly streamlines the manuscript's intention. We furthermore rephrased the abstract and introduction to clarify the focus of this paper.

Additionally, the section on the user survey has been moved to the supplementary material. We agree that the width and execution of the survey are slim for a standalone publication. Furthermore, the survey results are now presented in absolute numbers rather than percentages. The results of the user survey are now used as a supporting statement regarding the quality of the external documentation, code readability, and code modifiability in Section 6.

The abstract now reads (Lines 11-31):

"**Abstract**

*Global hydrological models (GHMs) improve our understanding of water flows and storage on the continents and have undergone significant advancements in process representation over the past four decades. However, as research questions and GHMs become increasingly complex, maintaining and enhancing existing model codes efficiently has become challenging. Issues such as non-modular design, inconsistent variable naming, insufficient documentation, lack of automated software testing suites, and containerization hinder the sustainability of GHM research software as well as the reproducibility of study results obtained with the help of GHMs. Although some GHMs have been reprogrammed to address these challenges, existing literature focuses on evaluating the quality of model output rather than the quality of the reprogrammed software. To address this research gap and guide other researchers who wish to implement their existing models as sustainable research software, we describe in detail how the most recent version of the GHM WaterGAP was reprogrammed. The reprogramming success is evaluated against numerous software sustainability criteria and the principles of findability, accessibility, interoperability, and reusability for research software (FAIR4RS), given that the objective of reprogramming was to enhance software sustainability and thus reproducibility of research results, as opposed to improving model output. Following an agile project management approach, WaterGAP was rewritten from scratch in Python with a modular Model-View-Controller architecture. Due to the switch from C/C++ in the legacy code to Python, execution time doubled. Our evaluation indicates that the reprogramming substantially improved the software's usability, maintainability, and extensibility, making the reprogrammed WaterGAP software much more sustainable than its predecessor. The reprogrammed WaterGAP software can be easily understood, applied, and enhanced by novice and experienced modellers and is suited for collaborative code development across diverse teams and locations, fostering the establishment of a community GHM. We outline six lessons learned from the reprogramming process concerning the sustainability-runtime trade-off, the applicability of the agile approach, software design patterns, variable naming, external documentation, and automation.*"

The  section of the introduction on the research objectives now reads (Section 1, Lines 75-89):

*"To address this research gap and support the reprogramming of other legacy software, this paper provides a detailed account of the reprogramming process of GHM WaterGAP (Döll et al., 2003; Müller Schmied et al., 2024) and the characteristics of the new software. Reprogramming aimed to enhance the software's sustainability for long-term research use by a broad community and to increase the reproducibility of the computational research performed with this model. The success of the reprogramming was assessed by comparing the legacy code to the reprogrammed version according to numerous specific sustainability criteria and FAIR4RS principles. It is important to note that our goal in reprogramming WaterGAP was not to improve the model output; the reprogrammed software was to result in the same model output as the latest WaterGAP version 2.2e (Müller Schmied et al., 2024).*

*The paper is structured as follows: Section 2 introduces the WaterGAP model and the legacy software. Sustainability criteria for research software and methods relevant to this study are presented in Section 3. After describing the reprogramming process in Section 4, we present the architecture and new features of the reprogrammed software in Section 5. In Section 6, we evaluate the new WaterGAP software against selected sustainability criteria and the FAIR4RS principles. We also demonstrate that the reprogrammed and legacy software yield very similar model outputs (Section 7) and share lessons learned for others undertaking similar efforts (Section 8). Our conclusions follow in Section 9."*

We also added a new table to the method section that more clearly communicates the metrics we utilized to assess the software sustainability of the reprogrammed and legacy research software (Section 3, Lines 148-150):

[revised manuscript text omitted]

**Minor remarks**
Independent of the direction chosen I have a few smaller remarks on the current version of the text:
- Line 61, add a citation to Wilkinson 2016 for FAIR.

We added a citation to Wilkinson 2016 for FAIR (Section 1, Lines 63-65).
"Additionally, applying FAIR (Findable, Accessible, Interoperable, and Reusable) principles for research software (FAIR4RS) improves research software reusability, reproducibility as well as transparency (Barker et al., 2022; Wilkinson et al., 2016)."

- Line 65: the unsuspecting reader might conclude that "to improve … … were reprogrammed" was done as part of the aforementioned eWatercycle project

Thank you for the comment. We have revised the section to better highlight the broader problem statement, rather than implying a direct connection with the eWatercycle initiative. The revised section (Section 1, Lines 69–73) now reads:
"*Efforts to improve comprehension, usage, maintenance, extension, and collaborative development have led to the reprogramming of several models, including the global land surface model CLASSIC (Melton et al., 2020) and GHMs such as HydroPy (Stacke and Hagemann, 2021) and PCR-GLOBWB (Sutanudjaja et als., 2018). However, the publications on these reprogrammed software focus on evaluating the performance of the model output and lack a detailed account of the reprogramming process and an evaluation of the success of the reprogramming effort*"

- Line 181 and continuing: I would avoid as much as possible using URLs as citations and use bibtex citations to websites (preferably with a "last accessed" field).

Thank you for the comment. We have replaced the majority of URLs in the revised manuscript as bibtex citations with a "last accessed" field.

- Section 4.1.2. I always learned that agile is a project management tool for when you have a fixed amount of hours, or people, to work on something, but the end goal is not fixed, just "create as much value as quickly as possible" (ideal for start-up culture). This is ideal when project goals are allowed to fluctuate during the project. In most scientific large projects the end goal is quite fixed (reprogram this model into python). Therefore, I would invite the authors to highlight why they settled on Agile as a project management practice for this project (there might well be very good reasons!)

Thank you for the valuable comment. We agree that Agile is often associated with projects where the end goal is flexible, which is common in start-up environments. However, we believe that certain Agile principles can also offer significant benefits in academic software development. Specifically, Agile supports flexibility in incorporating evolving research questions and enables effective progress tracking. Tools such as task boards and backlogs provide transparency and help manage workflows efficiently. This is particularly important in academic settings where timelines are often constrained and team composition can change frequently such as in PhD and Postdoc projects. Agile's emphasis on regular communication helps align the efforts of diverse contributors, including students,

researchers, and supervisors, ensuring everyone stays informed and coordinated throughout the project.

We now include this in the new lesson learned (Section 8, Lines 531-542)

" **8.2 Implementing an agile process benefits reprogramming also in an academic setting**

*The agile development process, along with the use of user stories, was essential to our reprogramming effort, enabling iterative improvements through continuous feedback. Agile principles offer significant benefits in academic software development. Specifically, Agile supports flexibility in incorporating evolving research questions and enables effective progress tracking. Tools such as task boards and backlogs provide transparency and help manage workflows efficiently. This is particularly important in academic settings where timelines are often constrained and team composition can change frequently, such as in PhD and Postdoc projects. Agile's emphasis on regular communication helps align the efforts of diverse contributors, including students, researchers, and supervisors (the "project owners"), ensuring everyone stays informed and coordinated throughout the project. User stories helped ensure that the software features matched the scientific requirements of WaterGAP. However, estimating the time needed to complete user stories was difficult, and coordinating an agile process with only a few developers in an academic setting was somewhat challenging. Despite this, we recommend this approach for other reprogramming projects as it supports timely and user-focused development and helps the team stay updated on progress.*

*"*

- Line 288: "senior developers" are introduced as a role, but not previously explained.

We have explained the term senior developers in the revised manuscript (Section 4, lines 189-191).

*"After the writing and approval of the project proposal, a preliminary meeting was held in November 2021 among six senior developers. These are late-stage PhDs, PostDocs, and Professors with extensive expertise in the WaterGAP model and are also actively involved in developing and maintaining the software."*

- Line 291: same for "software development advisor"

We have explained the term software development advisor in the revised manuscript (Section 4, lines 207-208).
*"The software development advisor guides the developers on best practices, architecture, and code quality to ensure robust and sustainable software. "*

**Concluding**
I really like the new python version of WaterGAP. (I like it so much that I would invite the authors to work together to add support for WaterGAP in eWaterCycle to make it even more accessible to hydrologists!). I hope the above suggestions will help in choosing a focus direction for the manuscript and emphasizing those parts throughout. I am happy to review an updated version of this manuscript.

Thank you for the thoughtful and encouraging feedback, and we are delighted to hear that you liked the Python version of WaterGAP. We appreciate your suggestion regarding integration with eWaterCycle. This is an exciting idea, and we would be very open to exploring such a collaboration in the future.

**Reply to Reviewer 2**

This paper presents the process and evaluation of re-programming of the legacy WaterGAP model into the new Python platform. While this study is of interest to the hydrological modeling community, the presentation of this manuscript needs to be substantively revised to highlight the scientific merit of the current study. In its current structure, I am afraid the manuscript does not meet the standard of scientific publication and cannot be accepted.

Thank you for the valuable feedback. The unclear objective and scope of the manuscript as well as its unsuitable structure were also mentioned by Reviewers 1 and 3. This is why we improve the manuscript in various aspects:

1) We have revised our abstract and introduction to more clearly communicate the manuscript's scope. The focus is not on presenting a new model version, with improved model output, but on presenting how the WaterGAP model was implemented in a new software that was to follow the principles of sustainable research software, including FAIR4RS. We are also scientifically investigating the value of reprogramming the scientific software by addressing the research question: "How can the software sustainability of a legacy scientific software be improved by reprogramming it?"

2) We revised the methods section to include a table that clearly explains the indicators used for evaluating software sustainability and FAIR principles.

3) We added a new "Lessons Learned" section based on the transition from C/C++ to Python, replacing the previous discussion section. Along with section 3 on software sustainability and FAIR principles, and section 4 on the reprogramming process, this strengthens the manuscript's focus on reporting software sustainability through reprogramming.

Please see our response to major points raised by reviewer 1 (to avoid repetition of lengthy text) for an extended discussion of our change on the revised abstract (Lines 11-31), revised section of the introduction (Section 1, Lines 75-89),  added table in method (Section 3, Lines 148-150), and the new lesson learned section (Section 8, Lines 497 - 565).

**Specific comments**

1. My major concern of this manuscript is its structure and organization. The manuscript contains sections with too many (10) main sections and levels of subsections. In addition, some subsections contain overlapping information (across different main sections). This tedious structure makes the general reading of the manuscript very difficult. In addition, Sections 5.1 and 5.2 contain subsections with identical titles (Architecture and new features), which should be avoided or simply merged together to reduce the level of subsections. In particular, Section 5.2.1 contains a single sentence with a reference to the Supplement and a GitHub link, which is completely unnecessary to make a separate subsection.

We appreciate the reviewer's feedback regarding the structure and organization of our manuscript. We have reduced the number of subsections and revised the manuscript throughout to convey a more concise message and improve readability. Furthermore, we moved the user survey section to the supplementary because the scope and depth of the survey do not justify a standalone section in the manuscript, as pointed out by Reviewer 1. This change reduces the number of main sections from ten to nine. We also reviewed the manuscript carefully to eliminate overlapping content. For example, Section 3.1, which initially described the software evaluation against sustainability criteria and FAIR principles, has been condensed into a concise table (see Table 1) that clearly summarizes how each criterion was assessed. This revision prevents redundancy with Section 4.3, which focuses on how these criteria were applied during software development. We also revised Section 5 by merging the previously separate subsections 5.1 and 5.2 (see line 359-427).

2. In general, the authors should either avoid using abbreviations in section titles, or spell out their full names. Examples including FAIR4RS (Section 3.1.3 and 6 title), WGHM (5.1), GWSWUSE (5.2).

We have replaced all abbreviations in the revised manuscript with their full names.

3. Section 3.1.1 "Indicators for best practices in software engineering": this section seemingly contains information that overlaps with Section 4.3.1 "Best practices for code development", such as document and version control and automation. Please consider re-structure these parts to make the presentation more precise by, e.g. merging redundant information.

Thank you for the valuable feedback. Please see our response to your specific comment 1.

4. Figure 2: the right panel of the figure contains major user stories that are purely text. Such information can be better presented using a table or in-text description rather than a graphic presentation (figure).

Thank you for the suggestion. We tested a table in a previous version of the manuscript and found that it is more challenging to connect stories to the chart shown in Figure 2 (left panel). Discussing all complete user stories in the main manuscript would increase the text size, which likely would impact the readability of the already extensive paper. We thus chose to keep the representation as is, as we believe that through the representation as one figure, readers will be able to identify and connect the major implemented user stories (right panel) with the cumulative plot showing the remaining user stories and the corresponding working hours (left panel). Together, this provides an overall picture of what was implemented during the reprogramming process. For the convenience of readers who would like to see a table, we included it in the supplementary material (progress_taking.xlsx) and now reference it in the figure description.

5. Section 4.3 title: get rid of the period after "Code development".

We have revised the said title accordingly.

6. Section 4.3.2: narrative in this section contains many hyperlinks to GitHub repositories, which impedes the flow of the text and makes reading difficult. Authors may consider documenting these links in supplementary information (SI) and making reference to SI.

Thank you for the comment. We have replaced all hyperlinks to GitHub repositories in revised manuscript as bibtex citations with a "last accessed" field.

7. Table 2: A graphic presentation (e.g. bar plots) is recommended to better show the difference between the outputs of the reprogrammed and legacy software.

Thank you for the suggestion, however, we believe that Table 3 (previously Table 2) already provides a clear and comprehensive summary of the differences in the global-scale water balance components (in km³ yr⁻¹) between the reprogrammed and legacy WaterGAP global hydrological models.

8. Sections 9 does not contain much in-depth discussion, but rather recap of what has been presented in previous sections. Consider either merging this into concluding remarks (Section 10).

Thank you for your valuable comment. In response, we have removed the previous discussion section and included a dedicated "Lessons Learned" section. We also refer you to our response to the major points raised by Reviewer 1, where we elaborate further on the "Lessons Learned" section.

**Reply to Reviewer 3**

This work focuses on the model development and updates to a legacy global hydrological model, after reprogramming (e.g., different structures and languages, from C/C++ to python), aiming to make this software more sustainable and accessible to model users. The reviewer considers that the authors have done sufficient efforts to develop the numerical model, while it is struggle to me is that whether this kind of article type or writing style is acceptable to the journal GMD. Several years ago, a popular storm surge modelling community suggests how to consider the modelling work aside from publication. While it is now in consistence that modeling work is as important as publications, my concern is that whether this style could be published in a scientific journal.

For a journal, the novelty is very important, and previous limitations should also be pointed out and summarized. Regarding this manuscript, one major concern to me is the structure, which includes overly 10 parts, and should be reduced significantly. This leads to a presentation of the work, instead of proposing an assumption, and solving it in a scientific way. Secondly, it should be clear what is the advantages of this new model, e.g., computational efficiency or stability? The reviewer considers that the model updates should first consider the accuracy, e.g., whether the new model improves the model simulation results. So, several case studies or model applications should be included, and model-to-data comparison of hydrological variables should be conducted and listed.

Thank you for your valuable feedback. According to our view, the scientific merit of our manuscript is in the detailed description of the re-programming of a legacy software into a sustainable research software that is to support other researchers who may need to reprogram other (modelling) software.

The innovation is not in an improved model output but in a more sustainable model software, which we rigorously evaluate in the manuscript.

We clearly state in our revised manuscript (Section 1, Line 80-82).
*"It is important to note that our goal in reprogramming WaterGAP was not to improve the model output; the reprogrammed software was to result in the same model output as the latest WaterGAP version 2.2e (Müller Schmied et al., 2024)."*

We strongly believe that a paper of the type we submitted will help advance computational research, in particular its efficiency and reproducibility, and is thus well-placed in the Journal Geoscientific Model Development. We improved our manuscript in various aspects:

1) We have revised the paper structure to reduce sections from 10 to 9 by moving the user survey to the supplementary material.
2) We have revised our abstract and introduction to clearly communicate the manuscript's objective.
3) We revised the methods section to include a table that clearly explains the indicators used for evaluating software sustainability and FAIR principles.
4) We added a new "Lessons Learned" section based on the transition from C/C++ to Python, replacing the previous discussion section. Along with section 3 on software sustainability and FAIR principles, and section 4 on the reprogramming process, this strengthens the manuscript's focus on reporting software sustainability through reprogramming.

For further details and explanations, we also refer you to our response to Reviewer 2's specific comment 1, where we elaborate on the revised paper's structure, as well as to our response to Reviewer 1, which highlights the overall intention and how the content aligns with that aim.

Here are additional detailed comments and suggestions:
1. L124-125: please provide reasons why this work only concerns the reprogramming of GWSWUSE and WGHM models operating at the 30 arc-minute resolution. Why do try it based on 5-arc-minute resolution?

We assume the reviewer asks us to explain why we do not increase the spatial resolution while reprogramming the model, since we discuss spatial resolutions in the first paragraph of the introduction. We added this explanation to section 2.1.

It now reads (Section 2, Lines 119-122):

*"This paper only concerns the reprogramming of the GWSWUSE and WGHM models operating at the 30 arc-minute resolution. The reprogrammed code, however, is flexible enough to also run at higher spatial resolutions if the appropriate inputs are supplied and processes specially tailored to the 30-arcminute resolutions are adapted (e.g., the snow processing routine and the water usage distribution to neighbouring cells)."*

2. Avoid using abbreviations in the subsection title, e.g., FAIR4RS, WGHM, GWSWUSE etc.

We have replaced all abbreviations in revised manuscript with their full names.

3. Some subsections are too short, and some even start with the subsection title without any contents, e.g., 5.1, 5.2 etc.

Thank you for the feedback. We have revised Section 5 by merging the previously separate subsections 5.1 and 5.2 into a single, unified section. We have also implemented further improvement for readability in response to reviewer 2.

4. In Table 2, what do the values in the bracket mean?

We have revised the heading of Table 2 (now Table 3) to clarify the meaning of the bracket or parentheses (section 7 line 480-481). The added section to the heading now reads "Values without parentheses correspond to the reprogrammed WGHM, while values in parentheses refer to the legacy WGHM"

5. Figure 5, the percentage is used for the survey results, so the reviewer suggests including the sample size or the total sample number.

Thank you for the valuable feedback. The section on the user survey and the corresponding result have been moved to the supplementary material. We agree with Reviewer 1 that the survey's width and execution are slim for a standalone publication. We have also added the sample size to the figure's caption, which is now shown in the supplement.

---

## Author Response (AR2)

Dear Ting Sun,

Thank you very much for handling our manuscript and for your feedback. We have addressed your feedback through:

1) Moving a subsection to the Appendix. This significantly improved the readability.
2) We further improved the phrasing in our conclusions to make them more readable as well.

We would also like to highlight that the questions raised by Reviewer 3 are due to their inability to access the revised manuscript, whereas Reviewer 1 had access.

For clarity, reviewer and editor comments are highlighted in blue, our responses are in black, and any newly added text appears in italics. All sections and line numbers refer to the revised version of the manuscript after implementing feedback from the minor revision.

With regards on behalf of all authors,

Emmanuel Nyenah

**Response to Editor**

The reviewers have now evaluated your responses. While most of the earlier concerns have been addressed satisfactorily, a few issues remain—particularly regarding the readability of the manuscript.

We have revised the manuscript and moved Section 7 (Differences between the outputs of the reprogrammed and legacy software) to the Appendix. It is now referenced in the main text under quality assurance (Section 4.6). This revision reduces the number of main sections from 9 (previously 10) to 8, thereby supporting our main conclusions while improving the overall readability of the manuscript.

The referenced text under section 4.6, Lines 358-360, reads:

"*Comparison of legacy and reprogrammed software output. We verified that the reprogrammed software produces similar outputs to the legacy software, as demonstrated in the global water balance and analysis of renewable water resources (see Appendix).* "

**Response to Reviewer 1: Rolf Hut**

I thank the authors for the thorough re-working of the manuscript. In this version, I am more than happy to recommend this for publishing.

Thank you once again for your feedback, which greatly helped improve the manuscript. We are glad to hear that you are now satisfied with the revised version.

**Response to Reviewer 3**

The authors have addressed my comments and suggestions overall well. This work could be published in GMD after further addressing my following comments and suggestions properly:

Thank you for the valuable feedback on our original contribution, which greatly helped us improve the manuscript.

1. I am not sure it is my handling technique of the website or other reasons I could not reach the updated or revised version of the manuscript. This makes it difficult to check whether the authors have made substantial revisions on the manuscript accordingly.

We appreciate your comment. However, we have substantially revised the manuscript, as also noted by Reviewer 1. These revisions addressed all previously raised concerns. To further improve the readability, as requested by the editor, we have moved Section 7 to the Appendix (see also Editor response).

2. Although the purpose of this manuscript is to give guidance or instructions for the model users to better employ this WaterGap software, the reviewer still considers that at least this update need validation to check the accuracy and reliability of this new model. For example, additional designed cases be included for simulations from the traditional and the new models, and observations are used to check the model performance for hydrological variables. At current version of the manuscript, it is difficult for the reviewer to judge that whether these new features could influence the model performance or not.

Thank you for the comment. Importantly, we have not introduced any new features with the software rewrite. Thus, the goal of this manuscript is not to evaluate the model's performance, but rather to examine the reprogramming process itself. Nevertheless, we assess whether the new model provides similar results to the original version (through a comparison of the water balance and spatial differences in long-term renewable water resources) and explain possible discrepancies.

The goal of the manuscript is not to provide guidance or instructions for model users to employ. These goals were clearly stated in the introduction section (Section 1, Lines 74-81) and the abstract (Lines 18-23) of the first and second revised manuscripts. We are convinced that our revised version conveys a clear message about the research gap we are addressing in this manuscript.

To reiterate, we define the goal of the manuscript as follows (Section 1, Lines 74-81):

"To address this research gap and support the reprogramming of other legacy software, this paper provides a detailed account of the reprogramming process of GHM WaterGAP (Döll et al., 2003; Müller Schmied et al., 2024) and the characteristics of the new software. Reprogramming aimed to enhance the software's sustainability for long-term research use by a broad community and to increase the reproducibility of the computational research performed with this model. The success of the reprogramming was assessed by comparing the legacy code to the reprogrammed version according to numerous specific sustainability criteria and FAIR4RS principles. It is important to note that our goal in reprogramming WaterGAP was not to improve the model output; the reprogrammed software was to result in the same model output as the latest WaterGAP version 2.2e (Müller Schmied et al., 2024)."

3. The structure. This manuscript describes a lot of aspects of the developed new model. For a scientific paper and publication, it is suggested that only the key findings and importance information should be included in the manuscript in a well-structured order. For example, the IMReD structure, i.e., Introduction, Method/Methodology, Results, Discussion, and Conclusion. The excessive will lead the reader to be unfocused on specific parts and lose the key information that this manuscript intends to present, so make the structure and organization of this manuscript more fluent and concise.

We appreciate the reviewer's feedback regarding the structure and organization of our manuscript. Please see the response to the editor where we elaborate on the new changes regarding the paper structure. We, however, believe that our original revisions to the paper have already improved the manuscript's structure.

Importantly, the paper, at its core, follows the IMReD structure (section 1 is the Introduction, section 2 is a somewhat traditional data or background section, section 3 is the methods, section 6 is the results, and section 7 is the discussion); however, we include additional sections to describe the programming process and software architecture (Sections 4 and 5). Additionally, instead of a traditional discussion section, we include a "Lessons Learned" section, which aligns with the paper's objective.

We guide readers through this structure as explained at the end of the Introduction (Section 1, Lines 83-87):

"The paper is structured as follows: Section 2 introduces the WaterGAP model and the legacy software. Sustainability criteria for research software and methods relevant to this study are presented in Section 3. After describing the reprogramming process in Section 4, we present the architecture and new features of the reprogrammed software in Section 5. In Section 6, we evaluate the new WaterGAP software against selected sustainability criteria and the FAIR4RS principles and share lessons learned for others undertaking similar efforts (Section 7). Our conclusions follow in Section 8 "

4. L124-124: If the model already runs at the 30 arc-minute resolution, the simulated results could be compared with the traditional model against observations, and also the 5 arc-minute resolution could be then done.

A model comparison against observations is already available in a recent publication (Müller Schmied et al 2024, see Section 7 Evaluation of WaterGAP v2.2e), where we compare streamflow and total water storage anomalies from the model (forced by various climate forcings) against observational data.

In the current paper, we are reprogramming the same model, WaterGAP 2.2e, as a sustainable research software. Including a separate model performance comparison would deviate from the paper's scope and significantly increase the length of an already extensive manuscript.

Additionally, the 5 arc-minute resolution version of the model has not yet been reprogrammed, and this paper only focuses on the 30 arc-minute resolution as stated in the manuscript (section 2, Lines 117-120).

**References**

Müller Schmied, H., Trautmann, T., Ackermann, S., Cáceres, D., Flörke, M., Gerdener, H., Kynast, E., Peiris, T. A., Schiebener, L., Schumacher, M., and Döll, P.: The global water resources and use model WaterGAP v2.2e: description and evaluation of modifications and new features, Geosci. Model Dev., 17, 8817–8852, https://doi.org/10.5194/gmd-17-8817-2024, 2024.

---

## Author Response (AR3)

Dear Ting Sun,

Thank you very much for handling our manuscript. We appreciate your prompt and helpful feedback. Below, we address each of your points in detail and outline the corresponding changes made to the manuscript. For clarity, your comments are highlighted in blue, our responses are in black, and any newly added text appears in italics. All sections and line numbers refer to the revised version of the manuscript.

With regards, on behalf of all authors,

Emmanuel Nyenah

**Response to Editor**

Thank you for submitting your work to GMD. I am pleased to inform you that your manuscript is acceptable for publication subject to minor revisions. Your work provides a valuable contribution documenting the transformation of WaterGAP into sustainable research software.

Thank you for highlighting the importance of our contribution to GMD with this manuscript.

However, two issues need attention:

1. Reference to Supplementary Materials

Replace all instances of "see Supplement" with proper citations:

- Archive supplementary files (software_requirement_WaterGAP.pdf, progress_taking.xlsx, etc.) in Zenodo

- Obtain DOIs

- Cite them properly in the manuscript

Thank you very much for spotting these issues. We agree that a generic reference to "see Supplement" is not helpful. We improved the manuscript in multiple places by providing more specific references, such as "see software specification document in the Supplement" or "see Excel file in the Supplement," to guide the reader more effectively.

We chose not to follow your suggestion to upload them to Zenodo because, according to the submission guidelines (https://www.geoscientific-model-development.net/submission.html), we are permitted to upload files smaller than 50 MB as supplementary materials, which will be assigned a DOI during publication by GMD. Our supplementary files mainly consist of supporting PDFs—such as the questionnaire, response documents, software requirements, and a table of user stories. We believe uploading to a different platform does not improve the manuscript and could instead confuse readers. Since we meet GMD's requirements for supplementary data, we would appreciate it if this choice also receives your final approval.

We further removed the "supplementary information for" title from our main supplemental file as stated in the guidelines as well.

2. Imbalanced Section 7 (Lessons Learned)

Currently imbalanced with lesson 7.1 at ~500 words while lessons 7.3-7.6 are only 50-80 words each. Please consolidate into 4 lessons:

- Keep: 7.1 Runtime trade-offs (already well-developed)

- Keep: 7.2 Agile process (expand to ~400 words with specific examples)

- Combine: 7.3 + 7.4 → "Code architecture and design practices" (~400 words)

- Combine: 7.5 + 7.6 → "Sustainable development practices" (~400 words)

Each lesson should include: challenge → solution → example → recommendation.

Thank you very much for the suggestion. We have revised Section 7.2 to include examples, and we have combined Sections 7.3 and 7.4, as well as Sections 7.5 and 7.6. We focused on ensuring that each lesson follows a clear structure as suggested, while remaining true to the lessons that resulted from our project.

The revised text for the new 7.2 to 7.4 now reads (Section 7, lines 508-565):

"

**7.2 Implementing an agile process benefits reprogramming also in an academic setting**

*The agile development process, along with the use of user stories, was essential to our reprogramming effort, enabling iterative improvements through continuous feedback. Agile principles offer significant benefits in academic software development. Specifically, Agile supports flexibility in incorporating evolving research questions and enables effective progress tracking. For example, tracking progress allowed us to monitor the number of user stories completed within each sprint, the time invested, and the remaining tasks to be tackled in the upcoming sprint. Such tracking helped us assess whether we were on track to complete the overall project within the required timeframe. Tools such as task boards and backlogs provide transparency and help manage workflows efficiently. This is particularly important in academic settings where timelines are often constrained and team composition can change frequently.*

*Agile's emphasis on regular communication helps align the efforts of diverse contributors, including students, researchers, and supervisors (the "project owners"), ensuring everyone stays informed and coordinated throughout the project. User stories helped ensure that the software features matched the scientific requirements of WaterGAP. Through sprint reviews and retrospective meetings (see Fig. 1), we collaboratively reviewed various user stories to assess whether changes in conceptualization and hence algorithms are needed. For example, we revised the algorithm governing surface water demand satisfaction due to the limited documentation available. During these meetings, we also discuss efficient technical solutions for some user stories, such as improving the runtime of the snow module and enhancing the overall runtime of the WGHM software. These discussions not only enabled conceptual alignment but also allowed us to find efficient technical solutions, incorporating input from product owners and the software development advisor. Despite these advantages, we faced several*

*challenges. Estimating the time needed to complete user stories proved difficult, and coordinating an agile process with only a few developers in an academic setting was somewhat challenging. Nevertheless, we recommend this approach for other reprogramming projects, as it supports timely and user-focused development and helps the team stay updated on progress.*

**7.3 Code architecture and design practices are paramount for readable, maintainable and modular software**

*Defining software architecture and its modular design is an iterative process that benefits greatly from the input of software experts. Architectural decisions play a critical role in determining how easily a model can be extended or modified without affecting other software components. For example, implementing each storage compartment as an independent Python module enabled targeted test development and comprehensive testing before integration. Guidance on software design patterns can be found, for example, in Gamma et al. (1994). A modular design also leads to improved readability, as single components of a project (e.g., code files) are more concise in their purpose.*

*Furthermore, good software engineering practice can further improve readability and maintainability, such as establishing meaningful and consistent variable names, which is also an iterative process that requires collaborative effort among developers and domain experts. For large projects, it is common for different developers to use various names for the same underlying concept, which can lead to confusion and subtle bugs if only one instance of a variable is updated (McConnell, 2004). For example, suppose a variable is referred to as both "storage_canopy" and "canopy_storage" in different parts of the code. In that case, an update to one variable may not automatically propagate to the other, resulting in inconsistencies in model output. Importantly, naming conventions need to be documented and enforced through the product owners and the active development team to guide future model development and reduce the risk of errors associated with ambiguous or inconsistent variable names.*

**7.4 Sustainable development practices such as documentation and automation ensures efficient software development and high software quality**

*Without sufficient documentation, it is challenging to comprehend the underlying concepts of algorithms and to modify, extend, or utilize the resulting software. An example is highlighted in Section 5, where we revised the algorithm governing surface water demand satisfaction and its impact on return flows to groundwater. The legacy code lacked sufficient in-code and external documentation, making it hard to understand and to re-implement. As a result, we developed an improved and consistent algorithm, accompanied by in-depth documentation. Throughout the project, we did not copy old documentation from the previous model; instead, we wrote it from scratch, keeping in mind the sustainability of the new software. We strongly recommend writing model documentation alongside code development rather than leaving it until the end. This approach helps capture critical assumptions, such as those embedded in algorithms, while they are still fresh in the developers' minds. Peer review of documentation improves its quality and clarity.*

*Manually updating documentation, running tests, and linting can be time-consuming, especially for large software projects. Developers may even forget to perform these tasks after modifying or extending the software, which can lead to buggy or broken code. Automating documentation generation reduces manual effort and helps keep the documentation up to date. Similarly, automating linting and testing ensures that the code functions correctly without the need for constant manual*

*checks. Automation is key to efficient development and high software quality, and we strongly recommend adopting this practice."*